# Suppression of heterotopic ossification in fibrodysplasia ossificans progressiva using AAV gene delivery

Yeon-Suk Yang[1,12], Jung-Min Kim [1,12], Jun Xie [2,3,4,12], Sachin Chaugule [1], Chujiao Lin[1], Hong Ma[2,3,4], Edward Hsiao [5], Jaehyoung Hong[6], Hyonho Chun[6], Eileen M. Shore [7,8,9], Frederick S. Kaplan[7,9,10], Guangping Gao [2,3,4,11] ✉ & Jae-Hyuck Shim [1,2,11] ✉

Heterotopic ossification is the most disabling feature of fibrodysplasia ossificans progressiva, an ultra-rare genetic disorder for which there is currently no prevention or treatment. Most patients with this disease harbor a heterozygous activating mutation (c.617 G > A;p.R206H) in ACVR1. Here, we identify recombinant AAV9 as the most effective serotype for transduction of the major cells-of-origin of heterotopic ossification. We use AAV9 delivery for gene replacement by expression of codon-optimized human ACVR1, ACVR1R206H allele-specific silencing by AAV-compatible artificial miRNA and a combination of gene replacement and silencing. In mouse skeletal cells harboring a conditional knock-in allele of human mutant ACVR1 and in patient-derived induced pluripotent stem cells, AAV gene therapy ablated aberrant Activin A signaling and chondrogenic and osteogenic differentiation. In Acvr1(R206H) knock-in mice treated locally in early adulthood or systemically at birth, trauma-induced endochondral bone formation was markedly reduced, while inflammation and fibroproliferative responses remained largely intact in the injured muscle. Remarkably, spontaneous heterotopic ossification also substantially decreased in in Acvr1(R206H) knock-in mice treated systemically at birth or in early adulthood. Collectively, we develop promising gene therapeutics that can prevent disabling heterotopic ossification in mice, supporting clinical translation to patients with fibrodysplasia ossificans progressiva.

Fibrodysplasia ossificans progressiva (FOP, #OMIM 135100) is an ultra-rare genetic disorder with a defining feature of progressive, disabling heterotopic ossification (HO) that forms within skeletal muscle, tendons, ligaments, fascia, and aponeuroses[1,2]. HO develops in childhood and early adulthood following episodic flare-ups that occur spontaneously or are triggered by minor trauma, injury, intramuscular injection, or inflammation[2]. The ectopic bone leads to progressive immobility and severe pain[2].

All patients with FOP have gain-of-function mutations in the bone morphogenetic protein (BMP) type I receptor *ACVR1*, and ~97% of FOP patients harbor a recurrent *ACVR1R206H* mutation (c.617G>A;p.R206H)[3,4]. Recent studies showed that *ACVR1R206H* confers dysregulated BMP-pSMAD1/5 signaling in response to Activin A, which normally inhibits BMP-Smad1/5 signaling through the wild-type (WT) ACVR1 receptor[5,6]. Notably, the therapeutic effect of anti-Activin A antibody appears specific to FOP, suggesting that Activin A acts as a ligand of the *ACVR1R206H* receptor[7,8]. Although multiple cell types leading to HO, such as Tie2+ endothelial cells[9], Scx1+ cells[10], and PDGFRα+Sca1+ fibroadipogenic progenitors (FAPs)[11], have been identified[12], FAPs appear to be the most consistent cell type and a major contributor to HO. However,

molecular mechanisms of such findings in coordination with a niche environment have yet to be fully elucidated.

FOP treatment is challenging due to the early age of onset of ectopic bone formation and the difficulty in selectively suppressing BMP signaling by the $ACVR1^{R206H}$ mutation. Presently, there is no definitive prevention or treatment for the progressively disabling HO except for symptomatic management with high-dose corticosteroids for episodic flare-ups, which can reduce symptoms like pain and edema[13]. Although a retinoic acid receptor γ agonist (palovarotene) was approved for treating FOP in Canada and an anti-Activin A antibody (REGN 2477), an immunosuppressant (rapamycin), and ACVR1 kinase inhibitors (IPN60130), are currently in clinical trials, use of these drugs is limited[14]. Given that gene therapy using recombinant adeno-associated viral (rAAV) vectors holds promise for treating many genetic disorders[15], AAV gene therapy may be a plausible therapeutic strategy for FOP. rAAVs have demonstrated high transduction efficiencies to skeletal muscle and the skeleton in vivo[16,17], long-term durability of therapeutic gene expression[18], and acceptable safety profiles in clinical studies. Of note, rAAVs have been evaluated in over 145 clinical trials and more than 2000 patients worldwide[19].

This study identifies AAV gene therapy as a promising therapeutic option for blocking HO in FOP. We demonstrate that the rAAV9 serotype effectively transduces the identified cells-of-origin of HO, FAPs in mice harboring a conditional knock-in allele of human $ACVR1^{R206H}$ ($Acvr1^{(R206H)KI}$). $Acvr1^{(R206H)KI}$ mice can be used to test the efficacy of AAV gene therapy on the skeletal and extraskeletal features of FOP as they model both the skeletal malformations and progressive HO found in human FOP patients[7]. Remarkably, AAV-mediated delivery of gene replacement, $ACVR1^{R206H}$ allele-specific silencing, or a combination of gene replacement and silencing was highly effective in suppressing both traumatic and spontaneous HO in these mice when administered transdermally (t.d.) or intravenously (i.v.) at birth or early adulthood. Mechanistically, AAV gene therapy ablated aberrant Activin A signaling and chondrogenic and osteogenic differentiation of human FOP patient-derived iPSCs and mouse $Acur1^{(R206H)KI}$ skeletal cells. Taken together, our proof-of-concept study using AAV gene therapy provides fundamental insights into a critical step toward clinical translation for FOP patients.

## Results

### AAV gene therapy targets the human $ACVR1^{R206H}$ receptor

FOP is caused by a heterozygous, activating mutation of $ACVR1$ (c.617G>A;p.R206H) in ~97% of patients[4]. Given recent evidence that wild-type ACVR1 ($ACVR1^{WT}$) blocks BMP signaling and leads to an inhibition of HO in FOP mice[11], we hypothesized that dilution of dysregulated BMP signaling of $ACVR1^{R206H}$ with overexpressed $ACVR1^{WT}$ might inhibit dysregulated BMP signaling and HO in FOP (hereafter, referred to gene replacement, Supplementary Fig. 1). To inhibit the dysregulated BMP signaling of human $ACVR1^{R206H}$, a codon-optimized version of human $ACVR1$ complementary DNA ($ACVR1^{opt}$, 74% nucleotide identity compared to wild-type coding sequence) was cloned into the AAV vector genome containing a CMV enhancer, chicken β-actin promoter (CBA), and intron. To reduce the size of the AAV vector, the 1027 bp CBA intron was replaced by a 384 bp Mass-Biologics (MBL) intron or a 172 bp synthetic intron (Fig. 1a), validating abundant expression of the $ACVR1^{opt}$ receptor driven by all three promoters (Fig. 1b). Since high levels of AAV-delivered shRNAs can perturb native RNAi machinery and cause off-target silencing effects[20,21], we developed AAV-compatible artificial miRNA (amiR) by inserting the guide strand of a small silencing RNA into a mouse miR-33-derived miRNA scaffold, which substantially improved conventional shRNA-related toxicity and off-target silencing[22]. Using a sequence walk methodology, 12 amiRs targeting human $ACVR1^{R206H}$ mRNA, but not human $ACVR1^{WT}$ and $ACVR1^{opt}$ mRNAs, were designed and inserted in the intron between the CBA promoter and the

mCherry reporter gene (Fig. 1c). Additionally, the sensor plasmids encoding firefly luciferase (Fluc), $Renilla$ luciferase (RLuc), and complimentary sequences of human $ACVR1^{R206H}$, $ACVR1^{WT}$, or $ACVR1^{opt}$ in 3'-UTR of RLuc, were generated to screen $ACVR1^{R206H}$ allele-specific amiRs (Supplementary Fig. 2a). RLuc activity was normalized to Fluc activity and lower luciferase activity represents higher silencing efficiency of amiRs (Fig. 1d). Moreover, mammalian expression vectors of human $ACVR1^{R206H}$, $ACVR1^{WT}$, or $ACVR1^{opt}$ were co-transfected with amiRs, and protein or mRNA levels of ACVR1 receptor were assessed (Fig. 1e, Supplementary Fig. 2b and c). Among the 12 amiRs, $amiR$-$RH6$, and $amiR$-$RH7$ were selected as the most effective gene silencers specific to $ACVR1^{R206H}$, with little to no silencing effect on $ACVR1^{WT}$ and $ACVR1^{opt}$. Finally, the combination of $amiR$-$RH6$ or $amiR$-$RH7$ and $ACVR1^{opt}$ cDNA was cloned into the AAV vector genome containing the CBA promoter with MBL or synthetic (Syn) intron (Fig. 1f). The knockdown efficiency of $ACVR1^{R206H}$ receptor by $amiR$-$RH6$ or $amiR$-$RH7$ (Fig. 1g), protein levels of $ACVR1^{opt}$ receptor (Fig. 1h), and the ability of these plasmids to inhibit Activin A-induced luciferase activity (Fig. 1i). Thus, AAV plasmids designed to reduce the dysregulated signaling by $ACVR1^{R206H}$ through gene replacement with $ACVR1^{opt}$ expression, through gene silencing by $ACVR1^{R206H}$ allele-specific amiRs, or through the combination (Supplementary Fig. 1) were successfully generated and their functions were validated in vitro.

### AAV gene therapy inhibits osteogenic potentials of human FOP iPSCs

To identify the optimal AAV serotype for transducing human and mouse osteogenic progenitors in vitro, a self-complementary AAV (scAAV) vector genome expressing the enhanced green fluorescent protein ($Egfp$) reporter gene was packaged into 15 conventional AAV capsids (rAAV1, rAAV2, rAAV2-TM[23], rAAV3b, rAAV4, rAAV5, rAAV6, rAAV6.2[24], rAAV7, rAAV8, rAAV9, rAAVrh8, rAAVrh10, rAAVrh39, and rAAVrh43)[25], and incubated with induced pluripotent stem cells (iPSCs) derived from dermal fibroblasts of human FOP patients[26,27], human bone marrow-derived mesenchymal stromal cells (BMSCs), human adipose-derived mesenchymal stromal cells (ASCs)[28,29], or mouse skeletal muscle progenitors (C2C12). Expression of EGFP in transduced cells was assessed by immunoblotting (Fig. 2a) and fluorescence microscopy (Supplementary Fig. 3). Four AAV serotypes, rAAV2, rAAV5, rAAV6, and rAAV6.2, were able to transduce all four cell types, while rAAV4 was only able to transduce ASCs and C2C12 cells, and rAAV9 was only able to transduce BMSCs. Since rAAV6.2 showed the highest transduction efficacy in all four cell types, the constructs expressing $amiR$-$RH6$ or $amiR$-$RH7$ and $ACVR1^{opt}$ were packaged into the AAV6.2 capsid, and their genome integrity was validated (Fig. 2b).

Since large amounts of tissue from FOP patients cannot be obtained due to the substantial risks of inducing more HO, human FOP iPSCs were generated. iPSCs[26,27,30] and dermal fibroblast-derived osteoblasts[31,32] have been well established for in vitro FOP modeling and for applications in drug screening. To test the effects of AAV gene therapy on human FOP iPSCs in vitro, cells were transduced with rAAV6.2 carrying $egfp$ (control), $amiR$-$RH6.ACVR1^{opt}$, or $amiR$-$RH7.ACVR1^{opt}$ (combination therapy). Silencing specificity of $amiR$-$RH6$ or -$RH7$ was examined by next-generation sequencing (NGS), demonstrating that while EGFP control-expressing FOP iPSCs displayed 65.4% $ACVR1^{R206H}$ vs. 34.6% $ACVR1^{WT}$ transcripts, the transcript pattern was substantially shifted to 36.7% vs. 63.3% ($amiR$-$RH6.ACVR1^{opt}$) and 39.5% vs. 60.5% ($amiR$-$RH7.ACVR1^{opt}$, Fig. 2c, top). Expression of $ACVR1^{opt}$ and EGFP was also confirmed in these cells (Fig. 2c, bottom, Supplementary Fig. 4a). Whole transcriptome analysis further demonstrated that compared to control, $amiR$-$RH6.ACVR1^{opt}$ and $amiR$-$RH7.ACVR1^{opt}$ induced upregulation of 45 and 140 genes and downregulation of 27 and 87 genes, respectively (Fig. 2d), whereas the numbers of differentially expressed genes were

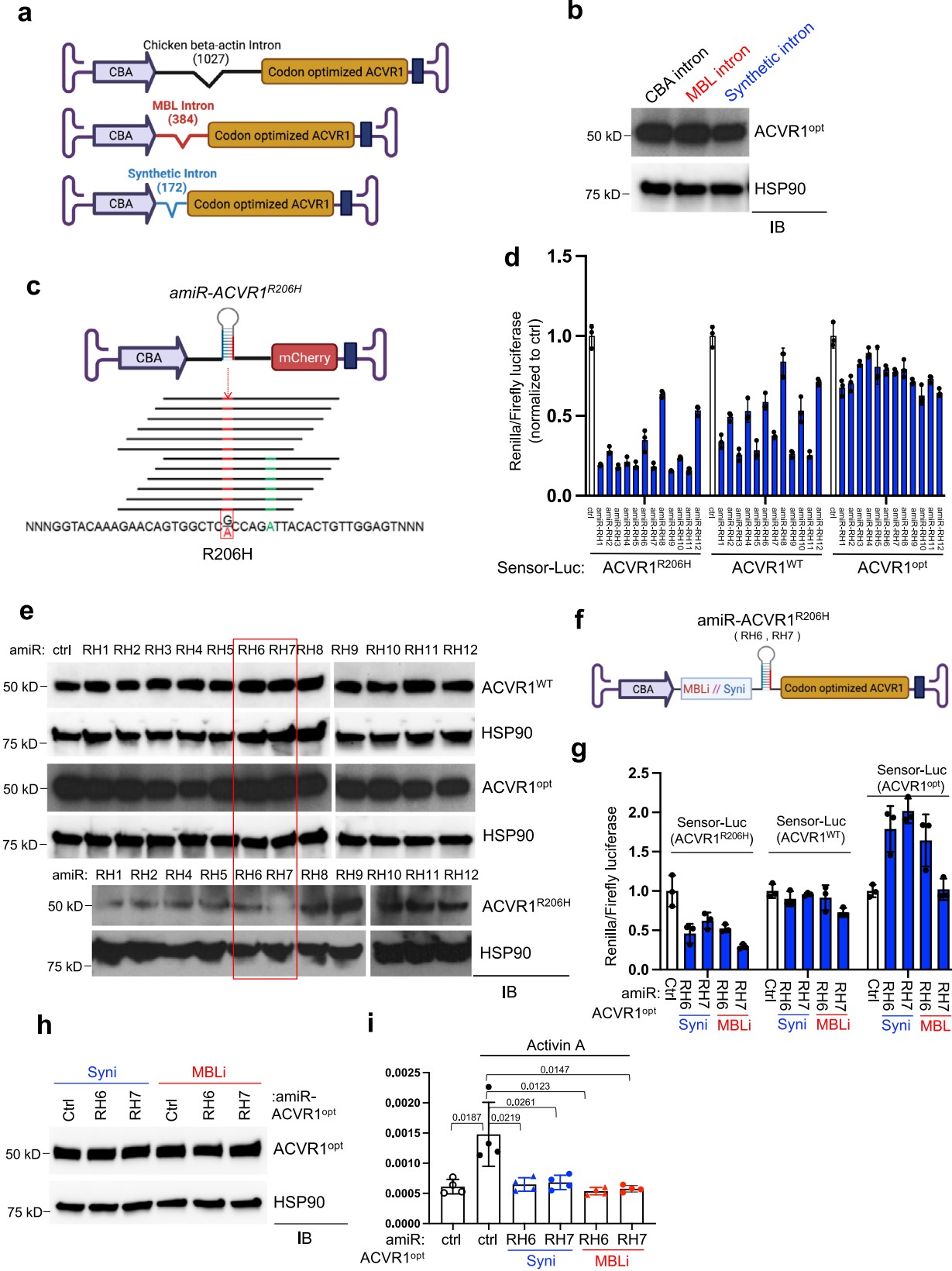

markedly reduced to 8 (upregulation) and 1 (downregulation) genes when comparing *amiR-RH6.ACVR1^opt* and *amiR-RH7.ACVR1^opt* (Supplementary Fig. 4b). These results suggest similar silencing specificity of *amiR-RH6* and *amiR-RH7*. Finally, we examined the ability of these AAVs to suppress osteogenic differentiation of FOP iPSCs and demonstrated a significant decrease in alkaline phosphatase (ALP)

activity (an early osteogenic marker), mineral deposition (a late osteogenic marker), and mRNA expression of *RUNX2* (a marker of early osteoblastic gene expression, Fig. 2e and f). Since only *amiR-RH6.ACVR1^opt* treatment did not alter ALP, mineralization, or *RUNX2* expression in human WT iPSCs generated from healthy donors, the therapeutic effects of *amiR-RH6.ACVR1^opt* might be more specific to

**Fig. 1 | Development of AAV vector targeting the human *ACVR1^R206H* receptor.**
**a** Schematic diagram of the plasmids expressing a codon-optimized version of the
human ACVR1 complementary DNA (*ACVR1^opt*) with the CBA promoter and three
different introns (created with biorender.com). CBA: CMV enhancer/chicken β-
actin promoter. **b** Validation of the expression of the ACVR1^opt receptor. Plasmids
expressing *ACVR1^opt* were transiently transfected into HEK293 cells and cell lysates
were subjected to immunoblotting with anti-ACVR1 antibody. Anti-HSP90 antibody
was used as a loading control. **c** Schematic diagram representing 12 amiRs that
target different sequence sites of human *ACVR1^R206H* mRNA (*amiR-ACVR1^R206H*).
The red box indicates the R206H mutation site (c.617G>A). Adenine (green) was mis-
matched to increase the selectivity of gene silencing (created with biorender.com).
**d** Plasmids encoding *amiR-ctrl* or 12 different *amiRs* were transiently transfected
into HEK293 cells along with amiR-sensor plasmids (sensor-Luc) that contain
*Renilla* luciferase and aimR complementary sequences for human ACVR1^R206H,
ACVR1^WT, and ACVR1^opt (*n* = 3). One day later, a luciferase assay was performed to
measure *Renilla* luciferase and normalized it to firefly luciferase. Lower activities
indicate higher silencing efficacy of amiRs. **e** Plasmids encoding *amiR-ctrl* or 12
different *amiRs* were transiently transfected into HEK293 cells along with a plasmid
expressing human ACVR1^R206H, ACVR1^WT, or ACVR1^opt cDNA and immunoblotted for
ACVR1. Anti-HSP90 antibody was used for loading control. **f** Schematic diagram of
the combination gene therapy constructs expressing *amiR-ACVR1^R206H* (RH6 or RH7)
and *ACVR1^opt* cDNA under the CBA promoter and MBL intron (MBLi) or synthetic
intron (Syni, created with biorender.com). **g, h** Plasmids encoding *amiR-ctrl* (ctrl),
Syni.amiR-RH6.ACVR1^opt, Syni.amiR-RH7.ACVR1^opt, MBLi.amiR-RH6.ACVR1^opt, or
MBLi.amiR-RH7.ACVR1^opt were transiently transfected into HEK293 cells along with
sensor-Luc plasmids. Luciferase assay (**g**, *n* = 3) or immunoblotting analysis for
ACVR1 (**h**) was performed. Anti-HSP90 antibody was used for loading control.
**i** Plasmids were transiently transfected into HEK293 cells along with the BMP
SMADs-responsive reporter gene (BRE-luc) and treated with Activin A (100 ng/ml,
*n* = 4). 24 h later, Activin A signaling activity was measured by luciferase assay. Data
are representative of three independent experiments (**b, e**). Values represent
mean ± SD by an unpaired two-tailed Student's *t*-test or one-way ANOVA test (**i**).

the human *ACVR1^R206H* mutation in iPSCs than those of *amiR-RH7.ACVR1^opt*.

Previous studies demonstrated that the *ACVR1^R206H* mutation
activates SMAD1/5-mediated BMP pathway signaling in response to
Activin A, whereas Activin A does not normally activate BMP-
pSMAD1/5 signaling via the ACVR1^WT receptor[5–7]. As expected, Activin
A treatment significantly upregulated the expression of BMP-
responsive genes, *ID1* and *MSX2*, in control-expressing FOP iPSCs,
but this induction was markedly reduced in the presence of *amiR-
RH6.ACVR1^opt* or *amiR-RH7.ACVR1^opt* (Fig. 2g, Supplementary Fig. 4c).
Thus, the combined approach of silencing *ACVR1^R206H* expression and
expressing the ACVR1^opt receptor is a potent inhibitor of Activin
A-induced aberrant BMP signaling and osteogenic differentiation in
human FOP iPSCs.

### AAV gene therapy suppresses dysregulated BMP pathway signaling in mouse *Acvr1^(R206H)KI* skeletal cells

Given that FAPs were identified as the cell-of-origin of HO in a mouse
model of FOP[11,12], a subset of FAPs was isolated from the skeletal muscle
of 4-week-old *Acvr1^(R206H)Fl*;*PDGFRα-cre* mice using cell surface markers
(PDGFRα⁺ScaI⁺CD31⁻CD45⁻)[11]. As seen in human FOP iPSCs, treatment
with *amiR-RH6.ACVR1^opt* or *amiR-RH7.ACVR1^opt* resulted in a significant
decrease in Activin A-induced ALP activity and osteogenic gene
expression (Fig. 2h and i). These results demonstrate therapeutic
effectiveness of combination treatment in both human and mouse
FOP cells.

To further define the potential therapeutic effects of *ACVR1^R206H*
allele-specific silencing, gene replacement, or the combination in FOP,
rAAV6.2 carrying *amiR-RH6*, *ACVR1^opt*, or *amiR-RH6.ACVR1^opt* were
transduced into mouse *Acvr1^(R206H)KI* osteogenic progenitors isolated
from *Acvr1^(R206H)Fl*;*PRRX1-cre* mice. Knockdown efficiency of *ACVR1^R206H*
and expression of *ACVR1^opt* were validated in these cells (Supplemen-
tary Fig. 5a). Since *amiR-RH6.ACVR1^opt* shows a higher specificity to the
human *ACVR1^R206H* mutation in iPSCs than *amiR-RH7.ACVR1^opt*
(Fig. 2d–f), *amiR-RH6.ACVR1^opt* and *amiR-RH6* were selected for further
studies. *amiR-RH6.ACVR1^opt* markedly reduced Activin A-induced
SMAD1/5 phosphorylation and *Id1* expression in *Acvr1^(R206H)KI* cells,
whereas little to mild reduction was detected in the presence of *amiR-
RH6* or *ACVR1^opt* alone, compared to EGFP control (Fig. 3a and b).
Notably, control-expressing *Acvr1^(R206H)Fl* (*Acvr1^WT*) cells were unre-
sponsive to Activin A. As a result, ALP and mineralization activities of
*Acvr1^(R206H)KI* osteogenic progenitors were reduced by *amiR-
RH6.ACVR1^opt* than *amiR-RH6* and *ACVR1^opt*, demonstrating inhibitory
effects of this combination strategy on both early and late osteogenic
differentiation (Fig. 3c, Supplementary Fig. 5b–d). This inhibitory
effect is specific to Activin A-induced osteogenesis of *Acvr1^(R206H)KI* cells
since BMP4-induced mineralization in these cells was not affected by
*amiR-RH6.ACVR1^opt* (Fig. 3d).

Next, we tested the therapeutic effects of AAV gene therapy on
Activin A-induced chondrogenesis of *Acvr1^(R206H)KI* chondrogenic pro-
genitors isolated from the knee joints of *Acvr1^(R206H)Fl*;*PRRX1-cre* neo-
nates at postnatal day 2 (P2). After AAV treatment, the knockdown
efficiency of *ACVR1^R206H* and expression of *ACVR1^opt* were validated in
these cells (Supplementary Fig. 5e). *amiR-RH6.ACVR1^opt* almost com-
pletely ablated Activin A-induced chondrogenesis, as shown by a sig-
nificant decrease in the expression of early (Sox9 and Type 2 Collagen)
and late (Aggrecan) chondrogenic genes and the production of carti-
lage matrix proteoglycans (alcian blue staining) (Fig. 3e, Supplemen-
tary Fig. 5f). Collectively, these results suggest that AAV-mediated
delivery of a combined *ACVR1^R206H* allele-specific amiR and *ACVR1^opt*
could potently suppress Activin A-induced aberrant BMP signaling,
osteogenesis, and chondrogenesis of human and mouse FOP cells,
with little to no effect on the ACVR1^WT receptor-mediated signaling and
BMP-induced signaling.

### Local delivery of AAV gene therapy suppresses traumatic HO in *Acvr1^(R206H)KI* FOP mice

Determining the transduction efficiencies of rAAVs in vitro is essential
to test different AAV-based strategies in relevant cell types; however,
the efficacy of an rAAV in vivo is impacted by multiple factors,
including the route of administration, serum factors, circulating neu-
tralizing antibodies, and multiple physiological barriers including the
ability to evade immune activation[25]. Therefore, to examine the ability
of rAAV6.2 to transduce the skeletal muscle where HO primarily
develops in FOP mice, rAAV6.2.*egfp* was administered intravenously
(i.v.) to mice, and EGFP expression in individual tissues was monitored
by IVIS-100 optical imaging (Supplementary Fig. 6a). As previously
reported[33], EGFP expression was only detected in the liver, whereas the
skeletal muscle (hindlimb) and heart showed weak expression when
i.v. administered together with vascular agents, including recombinant
vascular endothelial growth factor (VEGF)-166, sodium heparin, and
serum albumin (Supplementary Fig. 6b). Thus, rAAV6.2 was excluded
as a candidate serotype for in vivo FOP gene therapy due to its low
transduction efficiency in the skeletal muscle.

For our in vivo experiments, we turned to rAAV9, as this serotype
has been reported to be highly effective for transducing skeletal
muscle and bone[16,17] and it can transduce ~80% of muscle-resident
mesenchymal stem cells via intramuscular (i.m.) injection[34]. Our opti-
cal imaging data also confirmed a high transduction efficiency of i.v.
administered rAAV9 in the skeletal muscle (Supplementary Fig. 6c).
Since i.m. injection often induces muscle trauma in FOP patients,
causing HO lesions[35], skeletal muscle was transduced with rAAV9.*m-
Cherry* via transdermal (t.d.) injection using a hollow microneedle[36] to
minimize muscle trauma (Supplementary Fig. 6d and e). To examine its
ability to transduce HO-inducing FAP-lineage cells in the skeletal
muscle via t.d. injection, a mouse model representing the highest

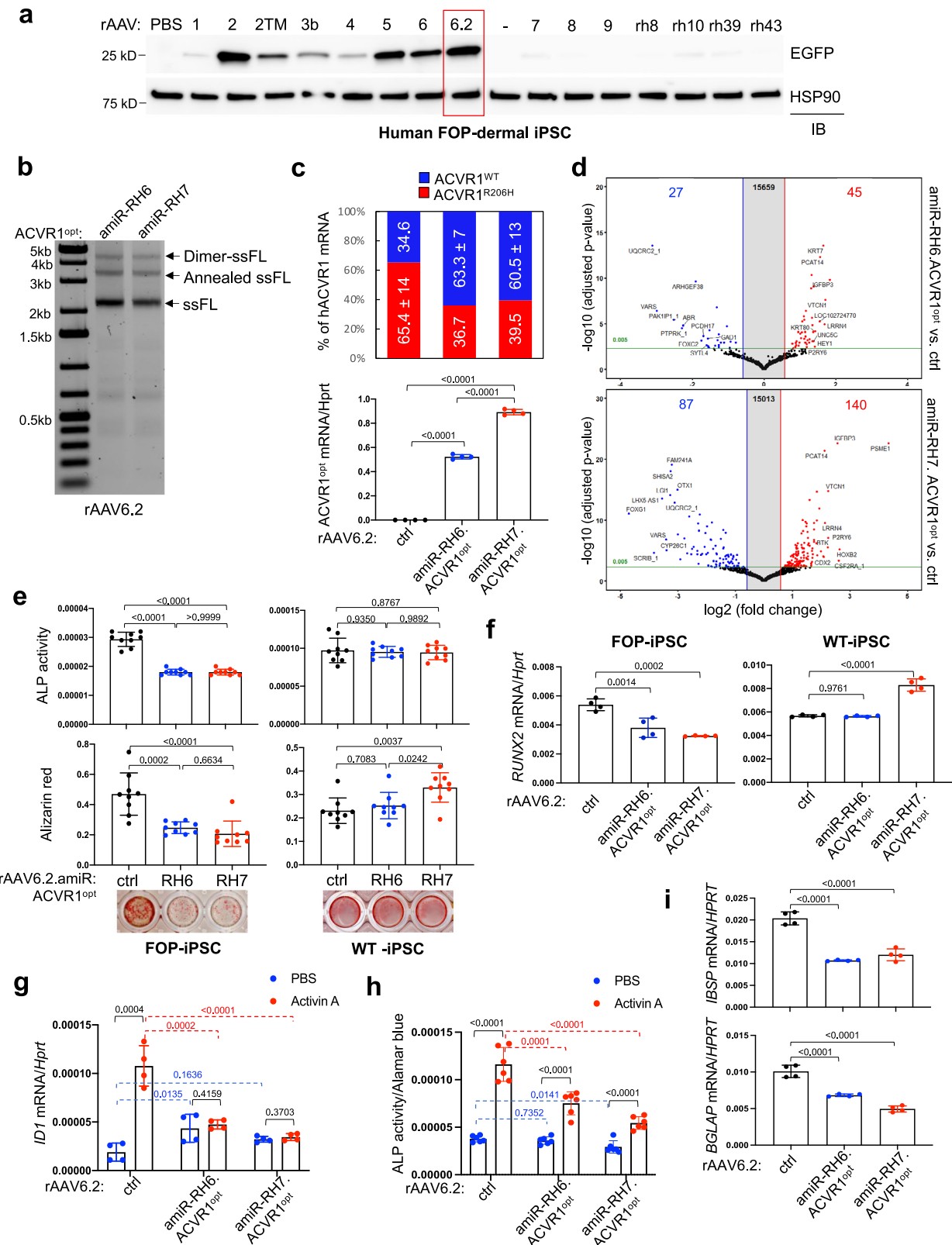

incidence forms of HO, muscle trauma/BMP-induced HO[37,38], was employed in *Tie2-cre;Rosa26^mCherry* reporter mice that express mCherry in a subset of FAP-lineage cells and endothelial cells[9,39]. One week after blunted muscle injury and i.m. administration of recombinant BMPs and matrigel into the quadricep of 2-month-old mice, rAAV9.*egfp* was injected t.d., and 3 weeks later, HO and EGFP expression were assessed by radiography and fluorescence microscopy, respectively (Supplementary Fig. 7a). In addition to the muscle fibers, a subset of Tie2+ FAP-lineage cells and osteoblasts within forming HO lesions also showed EGFP expression (Fig. 3f) while little to no expression was detected in the HO lesions treated with rAAV6.2.*egfp* (Supplementary Fig. 7b). These results demonstrated that t.d. delivery of rAAV9, not rAAV6.2, can transduce skeletal muscle cells and Tie2+ FAP- and osteoblast-lineage cells within forming HO lesions.

**Fig. 2 | Effects of AAV gene therapy in human FOP iPSCs and mouse *Acvr1*$^{(R206H)KI}$ cells. a** Human FOP iPSCs were treated with PBS or $5 \times 10^{10}$ genome copies (GCs) of 15 different AAV capsids packaged with the same *CBA-Egfp* transgene. 2 days later, EGFP expression was assessed by immunoblotting with an anti-GFP antibody. Anti-HSP90 antibody was used for loading control. **b** Genome integrity of rAAV6.2 carrying *amiR-RH6.ACVR1*$^{opt}$ or *amiR-RH7.ACVR1*$^{opt}$ was assessed by electrophoresis in the native gel. ssFL single-stranded full-length. **c, d** $5 \times 10^{10}$ GCs of rAAV6.2 carrying *EGFP control (ctrl)*, *amiR-RH6.ACVR1*$^{opt}$, or *amiR-RH7.ACVR1*$^{opt}$ were transduced to human FOP iPSCs, cultured under osteogenic conditions for 4 days, and subjected to next-generation sequencing (NGS) for ratio expression: *ACVR1*$^{R206H}$ vs. *ACVR1*$^{WT}$ (**c, top**) or RT-PCR for *ACVR1*$^{opt}$ expression (**c, bottom**, *n* = 4). Alternatively, total RNA was subjected to bulk RNA sequencing (**d**, *n* = 2). A volcano plot showing the gene expression for up/downregulated genes in the cells expressing *amiR-RH6.ACVR1*$^{opt}$ or *amiR-RH7.ACVR1*$^{opt}$ relative to *control*-expressing cells is displayed. A volcano plot was generated from multiple *t*-test.

**e–g** AAV-treated, human WT or FOP iPSCs were cultured under osteogenic conditions and alkaline phosphatase activity (ALP) and alizarin red staining were performed to assess early and late osteoblast differentiation, respectively (**e**, *n* = 9). Alternatively, osteogenic gene expression (RUNX2) was assessed by RT-PCR (**f**, *n* = 4). AAV-treated, human FOP iPSCs were incubated with PBS or Activin A (100 ng/ml) for 6 h, and *ID1* mRNA levels were measured by RT-PCR (**g**, *n* = 4). **h, i** PDGFRα⁺Sca1⁺CD31⁻CD45⁻ FAPs were sorted by FACS from the digested skeletal muscle of 4-week-old *Acvr1*$^{(R206H)Fl}$*;PDGFR*α-cre mice and transduced with $5 \times 10^{10}$ GCs of AAV6.2 carrying *EGFP control, amiR-RH6.ACVR1*$^{opt}$, or *amiR-RH7.ACVR1*$^{opt}$. 2 days later, AAV-treated FAPs were cultured under osteogenic conditions with PBS or Activin A (50 ng/ml) for 6 days, and ALP activity (**h**, *n* = 6) and osteogenic gene expression (BGLAP, IBSP, **i**, *n* = 4) were assessed for osteoblast differentiation. Data are representative of three independent experiments (**a, b**). Values represent mean ± SD by an unpaired two-tailed Student's *t*-test or one-way ANOVA test (**e–i**).

Since constitutive expression of the human *Acvr1*$^{R206H}$ allele results in perinatal lethality in mice[40,41], mice harboring a conditional *Acvr1*$^{R206H}$ knock-in allele (*Acvr1*$^{(R206H)Fl}$, Supplementary Fig. 7c) were crossed with Cre-ER$^{T2}$ mice (*Acvr1*$^{(R206H)Fl}$*;Cre-ER$^{T2}$*) where tamoxifen-induced expression of Cre recombinase mediates *Acvr1*$^{R206H}$-driven HO in early adulthood. Traumatic HO was induced by tibial muscle injury in these mice four weeks after t.d. injection of rAAV9 expressing the Cre recombinase (Supplementary Fig. 7d). Cre-mediated expression of *Acvr1*$^{(R206H)KI}$ in the gastrocnemius muscle resulted in HO following pinch injury/cardiotoxin injection (Fig. 3g and h), confirming the effectiveness of t.d. delivery of rAAV9 to HO-inducing cells in skeletal muscle.

To examine the ability of AAV gene therapy to suppress trauma-induced HO in FOP mice, 6-week-old *Acvr1*$^{(R206H)Fl}$*;Cre-ER$^{T2}$ mice were treated with tamoxifen to induce Cre recombinase expression, followed by t.d. injection of rAAV9 carrying EGFP control, *amiR-RH6*, *ACVR1*$^{opt}$, or *amiR-RH6.ACVR1*$^{opt}$. Three days later, pinch injury/cardiotoxin injection was introduced into the gastrocnemius muscle (Supplementary Fig. 7e). Knockdown efficiency of *ACVR1*$^{R206H}$ or *ACVR1*$^{opt}$ expression in AAV-treated gastrocnemius muscle was validated 4 weeks after muscle injury (Supplementary Fig. 7f). AAV treatment significantly reduced HO in the gastrocnemius muscle (Fig. 3i), demonstrating that local delivery of *ACVR1*$^{R206H}$ allele-specific silencing by *amiR-RH6*, gene replacement by *ACVR1*$^{opt}$ expression, and the combination of both was all effective in suppressing trauma-induced HO in the skeletal muscle of FOP mice. Notably, as BMP4-induced osteogenesis of *Acvr1*$^{(R206H)KI}$ osteogenic progenitors treated with *amiR-RH6.ACVR1*$^{opt}$ was largely intact (Fig. 3d), t.d. delivery of *amiR-RH6.ACVR1*$^{opt}$ did not affect muscle trauma/BMP-induced HO (Fig. 3j), suggesting the therapeutic effectiveness of rAAV9.*amiR-RH6.ACVR1*$^{opt}$ is specific to *ACVR1*$^{R206H}$-mediated genetic HO. Thus, AAV gene therapy is likely to act by inhibiting ACVR1$^{R206H}$-induced aberrant BMP pathway signaling and resultant chondrogenesis and osteogenesis in the skeletal muscle of *Acvr1*$^{(R206H)KI}$ mice.

### Systemic delivery of AAV gene therapy at birth prevents traumatic HO in *Acvr1*$^{(R206H)KI}$ FOP mice

All FOP patients with the classic *ACVR1*$^{R206H}$ mutation present with great toe malformations at birth and experience episodic flare-ups following minor trauma and HO in childhood or early adulthood[13]. We examined the ability of AAV gene therapy given at birth to prevent trauma-induced HO in adult FOP mice. To examine whether systemic delivery of rAAV9 at birth can transduce HO-inducing FAP-lineage cells in the skeletal muscle, a single dose of rAAV9.*mCherry* was injected i.v. into P1 PDGFRα-GFP reporter neonates[42] (Supplementary Fig. 8a), demonstrating mCherry expression in a subset of PDGFRα-GFP⁺ FAP-lineage cells in the skeletal muscle and PDGFRα-GFP⁺ osteoblasts and osteocytes in the trabecular and cortical bone compartments in addition to the heart, lung, liver, and kidney (Fig. 4a, Supplementary Fig. 8b and c).

These results were also confirmed in WT mice treated with rAAV9 expressing β-galactosidase (rAAV9.LacZ, Supplementary Fig. 8d). While a subset of chondrocytes in the articular cartilage show only GFP expression, there is no expression of GFP and mCherry proteins in the growth plate. These results suggest that a subset of chondrocyte-lineage cells in the articular cartilage, not in the growth plate, are positive for PDGFRα expression, but both of these are not rAAV9-transducible (Supplementary Fig. 8c). Thus, systemic delivery of rAAV9 at birth can transduce FAP-lineage cells in the skeletal muscle as well as osteoblast-lineage cells in the bone.

Next, a single dose of rAAV9 carrying EGFP control, *amiR-RH6*, *ACVR1*$^{opt}$, or *amiR-RH6.ACVR1*$^{opt}$ was administered i.v.to P1 *ACVR1*$^{(R206H)Fl}$*;Cre-ER$^{T2}$ neonates, and six weeks later, pinch injury/cardiotoxin injection was introduced to the gastrocnemius muscle 3 days after tamoxifen treatment (Supplementary Fig. 8e). Four weeks later, knockdown efficiency of *ACVR1*$^{R206H}$ and *ACVR1*$^{opt}$ expression in the gastrocnemius muscle was validated by RT-PCR analysis (Fig. 4b). Remarkably, *amiR-RH6.ACVR1*$^{opt}$ ablated the development of heterotopic bone and chondrogenic anlagen in the skeletal muscle while inducing nearly complete regeneration and reestablishment of normal muscle architecture. HO was markedly decreased in the presence of *amiR-RH6* or *ACVR1*$^{opt}$, but the therapeutic effect of *amiR-RH6* was highly variable compared to that of *amiR-RH6.ACVR1*$^{opt}$ and *ACVR1*$^{opt}$ (Fig. 4c and d). These results demonstrated that systemic delivery of AAV gene therapy at birth could prevent trauma-induced HO in *Acvr1*$^{(R206H)KI}$ FOP mice and that combination gene therapy is more effective than *ACVR1*$^{opt}$ expression or *ACVR1*$^{R206H}$ allele-specific silencing alone.

Muscle injury in a mouse model of FOP has been reported to induce the sequential pathological changes in HO lesion progression, including perivascular immune cell infiltration (Day 1–3), muscle degeneration and fibroproliferative response (Day 3–7), chondrogenesis (Day 7–14), and osteogenesis with heterotopic bone marrow establishment (Day 14–28)[9,43,44]. To define the stage at which *amiR-RH6.ACVR1*$^{opt}$ acts in HO pathogenesis, radiography, and histopathological evaluation of AAV-treated, the injured muscle was performed at various time points. 6 weeks after i.v. injection into *Acvr1*$^{(R206H)Fl}$*;Cre-ER$^{T2}$ neonates at P1, pinch injury/cardiotoxin injection was introduced to the gastrocnemius muscle 3 days post-injection of tamoxifen and HO pathogenesis was assessed at day 3, 7, 14, and 28 after muscle injury (Supplementary Fig. 8f). As expected, positive HO control muscle from EGFP control-treated *Acvr1*$^{(R206H)KI}$ FOP mice showed early injury responses at day 3, including muscle degeneration, immune cell infiltration, and fibroblast proliferation, while chondrogenic anlagen appeared at day 7, and began to transform into the heterotopic bone with bone marrow at day 14. HO was fully developed at day 28 post-injury (Fig. 4d–f). By contrast, there was little to no cartilage or heterotopic bone formation in *amiR-RH6.ACVR1*$^{opt}$-treated muscle and fibroproliferative responses were markedly reduced at day 14,

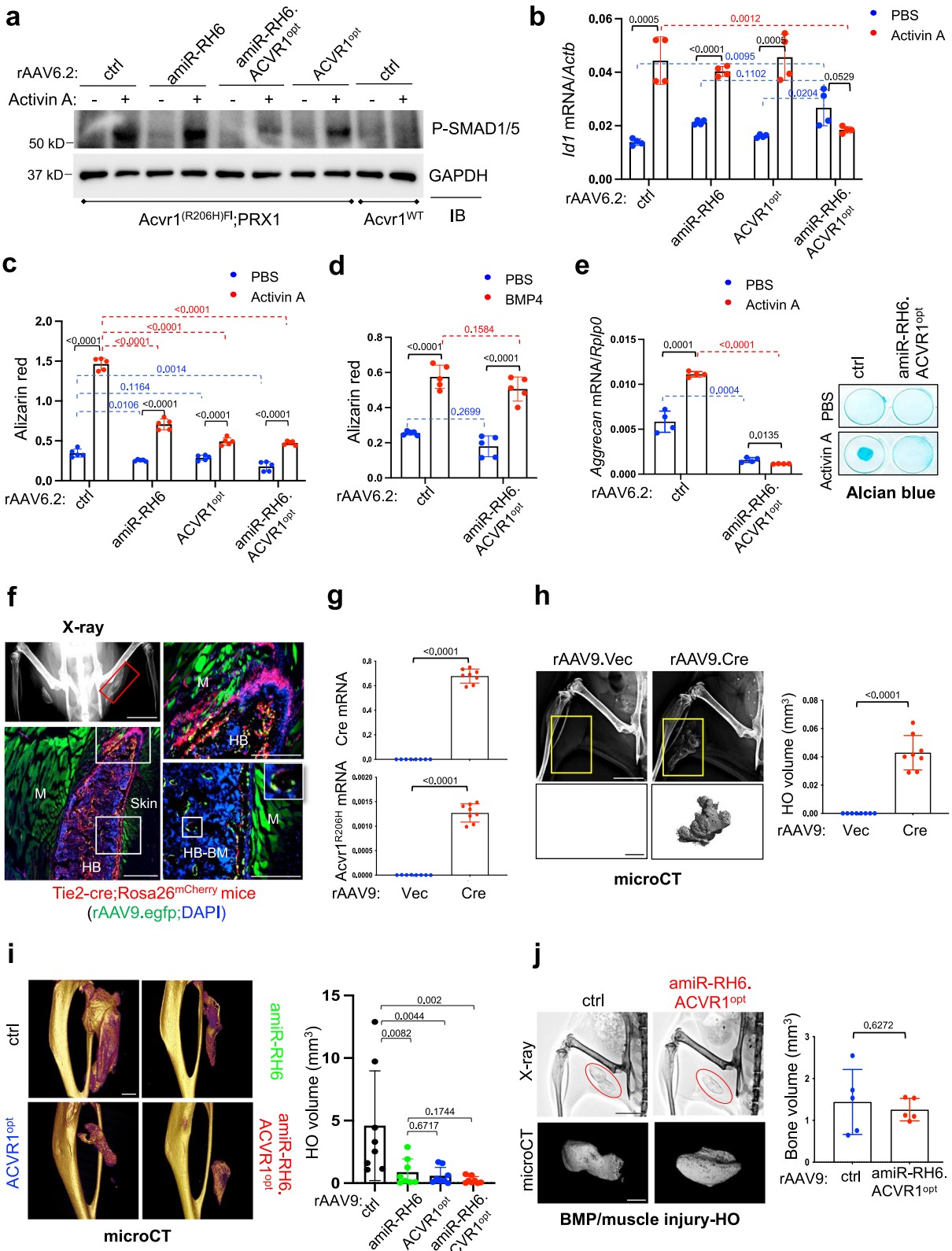

resulting in the nearly complete reestablishment of normal muscle architecture at day 28 (Fig. 4d–f). This corresponds to reduced levels of phosphorylated SMAD1/5 in the skeletal muscles treated with *amiR-RH6.ACVR1opt* relative to control, suggesting a potent inhibitory effect of *amiR-RH6.ACVR1opt* on aberrant activation of BMP signaling within the injured muscle of FOP mice (Fig. 4f). Notably, early injury responses at day 3, including infiltration of inflammatory

macrophages/monocytes (F4/80) and mast cells (toluidine blue), muscle degeneration, and fibroproliferation, were relatively comparable between the skeletal muscles treated with control and *amiR-RH6.ACVR1opt* (Fig. 4f). Likewise, 84 genes associated with inflammatory responses showed comparable expression patterns between control- and *amiR-RH6.ACVR1opt*-treated muscles except for *Tnfsf13/April, IL-11*, and *Ccl22* (Fig. 4g, h). Tnfsf13/April is a key cytokine that promotes B

**Fig. 3 | AAV gene therapy suppresses Activin A signaling and trauma-induced HO. a–d** *PRRX1*⁺ osteogenic progenitors isolated from 4-week-old *PRRX1-cre* (*Acvr1*^WT) or *Acvr1*^(R206H)Fl;*PRRX1-cre* femurs were transduced with $5 \times 10^{10}$ GCs of AAV6.2, stimulated with Activin A for 30 min, and immunoblotted for phospho-SMAD1/5. Anti-GAPDH antibody was used for loading control (**a**). *Id1* expression was assessed 6 h after Activin A stimulation (**b**, $n = 4$). AAV-treated cells were cultured under osteogenic conditions with PBS, Activin A (**c**, $n = 5$), or BMP4 (**d**, $n = 5$), and Alizarin red staining was performed. **e** *PRRX1*⁺ chondrogenic progenitors isolated from P2 *Acvr1*^(R206H)Fl;*PRRX1-cre* neonates were transduced with AAV6.2, and cultured under chondrogenic conditions for 4 or 6 days. *Aggrecan* mRNA levels or Alcian blue staining were performed ($n = 4$). **f** rAAV9.*egfp* was t.d. injected into the quadricep of 2-month-old male *Tie2-cre;Rosa26*^mCherry mice (red, $n = 3$) 1 week after i.m. injection with rBMP2/7/matrigel and muscle injury. 3 weeks later, radiography and histology of HO tissues were performed. The red box indicates a heterotopic bone (HB). M, muscle; HB-BM, heterotopic bone-bone marrow. Scale bars: 5 mm,

**left top**; 500 μm, **left bottom**; 400 μm, **right. g, h** rAAV9 was t.d. injected into the hindlimb of 6-week-old male *Acvr1R*^(R206H) mice ($n = 8$), and muscle injury was introduced 3 days post-injection. 4 weeks later, *ACVR1*^R206H and *Cre* recombinase expression was assessed by RT-PCR (**g**, $n = 8$) and HO was detected by radiography and quantified by microCT (**h**, $n = 8$). Scale bars: 5 mm, **left top**; 1 mm, **left bottom. i** rAAV9 was t.d. injected into the hindlimbs of 6-week-old female *Acvr1R*^(R206H)Fl;*Cre-ER*^T2 mice ($n = 8$) 3 days post-injection of tamoxifen. Muscle injury was applied 3 days post-injection. 4 weeks later, HO was assessed by microCT. Scale bar: 1 mm. **j** rAAV9 was t.d. injected into the quadricep of 2-month-old male WT mice ($n = 5$) followed by an rBMP2/7/matrigel injection and a muscle injury. 4 weeks later, HO was assessed by radiography and microCT. Scale bars: 5 mm, **top**; 1 mm, **bottom.** Data are representative of three independent experiments (**a**). Values represent mean ± SD by an unpaired two-tailed Student's *t*-test or one-way ANOVA test (**b–e**, **g–j**).

---

cell development by protecting from apoptosis[45], IL-11 is an IL-6 family member that contributes to hematopoiesis, bone development, tissue repair, and tumor development[46], and Ccl22 is a macrophage-derived chemokine that recruits TH2 cells into the inflammatory sites and the regulation of TH2-related immune responses[47]. Further studies will be necessary to define the contribution of these factors to trauma-induced HO in FOP mice. Consistent with our in vitro data showing the ability of *amiR-RH6.ACVR1*^opt to suppress Activin A-induced aberrant BMP signaling, chondrogenesis, and osteogenesis of human FOP iPSCs (Fig. 2e–g) and mouse *Acvr1*^(R206H)KI skeletal progenitors (Figs. 2h and I, 3a–e), these results demonstrate that i.v. administration of rAAV9.*amiR-RH6.ACVR1*^opt at birth prevented trauma-induced development of heterotopic endochondral ossification in *Acvr1*^(R206H)KI FOP mice but showed minimal effects on early post-traumatic injury responses. Thus, systemic delivery of combination AAV gene therapy at birth is a promising approach to preventing trauma-induced HO in FOP mice in early adulthood. Given previous and our own studies[16,17,48] showing cell-specific transduction of rAAV9 serotype in PDGFRα⁺ or Tie2⁺ FAP-lineage cells, osteoblast-lineage cells (osteoprogenitors, mature osteoblasts, osteocytes), and myoblast-lineage cells (myogenic progenitors, myoblasts, myocytes), but not fibroblasts and immune cells (monocytes, macrophages, dendritic cells, neutrophils, and T and B lymphocytes), systemically delivered *amiR-RH6.ACVR1opt* mainly impacts rAAV9-transducible cells, including FAP-, osteoblast-, and myoblast-lineage cells in injured sites, resulting in suppression of chondrogenesis and osteogenesis while facilitating muscle regeneration. However, *amiR-RH6.ACVR1opt* is likely to be ineffective for trauma-induced inflammation, fibroproliferation, and muscle damage (early post-traumatic injury responses) due to low transduction efficiency to fibroblasts and immune cells.

## AAV gene therapy given at birth prevents spontaneous HO in juvenile *Acvr1*^(R206H)KI FOP mice

Since PDGFRα⁺ FAP-lineage cells in the skeletal muscle interstitium are considered a major cell of origin thought to be responsible for HO[9,39], mice harboring a conditional knock-in allele of *Acvr1*^(R206H)Fl were crossed with PDGFRα-cre mice (*Acvr1*^(R206H)Fl;*PDGFRα-cre*), resulting in early-onset and widely distributed HO phenotypes of the musculature, tendons, and ligaments at multiple anatomical locations, including cervical spine, jaw, forelimb, hindlimb, hip, and ankle (Supplementary Fig. 9a)[11]. 3-week-old *Acvr1*^(R206H)Fl;*PDGFRα-cre* mice were i.v. injected with rAAV9 expressing LacZ and β-galactosidase expression in multiple HO lesions was assessed by histology 2 weeks post-injection. These results confirmed the effectiveness of systemically delivered rAAV9 to transduce HO-residing cells in the skeletal muscle, ligament, and tendon throughout the body (Fig. 5a, Supplementary Fig. 9b). We next tested whether a single dose of i.v. administration with the rAAV9 at birth can prevent spontaneous HO during skeletal development. P1 *Acvr1*^(R206H)Fl;*PDGFRα-cre* neonates were i.v. injected with rAAV9 carrying EGFP

control, *amiR-RH6*, *ACVR1*^opt, or *amiR-RH6.ACVR1*^opt and a full phenotypic characterization of these mice, including the natural history of progressive HO, bone remodeling, rate of disease progression, and survival rate, was performed. Compared to control-treated WT littermate control mice (*Acvr1*^WT;*ctrl*) showing a 100% survival rate, control-treated *Acvr1*^(R206H)Fl;*PDGFRα-cre* mice (*Acvr1*^R206H;*ctrl*) displayed a significant reduction in survival rate: 19% at 5 weeks and 100% lethality by 8 weeks (Fig. 5b). This survival rate was substantially improved by treatment with *amiR-RH6.ACVR1*^opt: 93% at 5 weeks, 67% at 8 weeks. Treatment with *amiR-RH6* or *ACVR1*^opt also increased survival rate: 50% or 67% at 5 weeks, 17% or 25% at 8 weeks, respectively. Likewise, control-treated FOP mice weighed an average of 59% less than littermate control WT mice, and weight loss was markedly ameliorated by the treatment with *amiR-RH6.ACVR1*^opt relative to *amiR-RH6* or *ACVR1*^opt alone (Fig. 5c). Intriguingly, the majority of control-treated FOP mice developed HO bilaterally at the temporomandibular joints (TMJ) by the age of 7 weeks old, failed to open their mouths, and died early due to starvation (Fig. 5d and e, Supplementary Fig. 10a), suggesting that jaw ankylosis is the primary reason for the reduced survival rate and substantial weight loss of *Acvr1*^(R206H)Fl;*PDGFRα-cre* mice. These phenotypes were also significantly ameliorated by treatment with *amiR-RH6.ACVR1*^opt while only mild improvement was seen in mice treated with *amiR-RH6* or *ACVR1*^opt alone (Fig. 5b–e). Thus, systemic delivery of *amiR-RH6.ACVR1*^opt at birth, not *amiR-RH6* or *ACVR1*^opt alone, almost completely prevented early-onset, spontaneous HO at the TMJs in juvenile FOP mice, resulting in an increase in survival rate and body weight.

Since osteoporosis is a clinical feature in many advanced FOP patients[13,49], bone mass and architecture in the lumbar vertebrae (L4) of AAV-treated *Acvr1*^(R206H)Fl;*PDGFRα-cre* mice were assessed by microCT and histology (Fig. 5d and f, Supplementary Fig. 10b). Control-treated FOP mice showed ~70% decrease in vertebral bone mass compared to littermate control WT mice, and this bone loss was prevented by systemic delivery of *amiR-RH6.ACVR1*^opt at birth, but not *amiR-RH6* or *ACVR1*^opt alone. Notably, fluorescence microscopy of *PDGFRα-EGFP* reporter mice demonstrated high expression of PDGFRα in osteoblasts and osteocytes within alveolar bone and dental pulp mesenchymal stem cells and odontoblasts within teeth (Supplementary Fig. 10c)[50]. Control-treated *Acvr1*^(R206H)Fl;*PDGFRα-cre* mice displayed low alveolar bone mass in both maxillary and mandibular bones. However, tooth morphology and dentin mass were largely normal in these mice, suggesting that the *ACVR1*^R206H mutation may not directly affect tooth development. Similar to the vertebral bone, alveolar bone loss in FOP mice was also almost completely prevented by treatment with *amiR-RH6.ACVR1*^opt (Fig. 5g, Supplementary Fig. 10d and e). Further studies will be necessary to define the contribution of the *ACVR1*^R206H mutation to this process using a tooth-specific Cre mouse line[51].

As previously reported[11,52], whole body microCT scanning and radiography of control-treated, 5-week-old *Acvr1*^(R206H)Fl;*PDGFRα* mice showed spontaneous HO at multiple anatomical locations, including

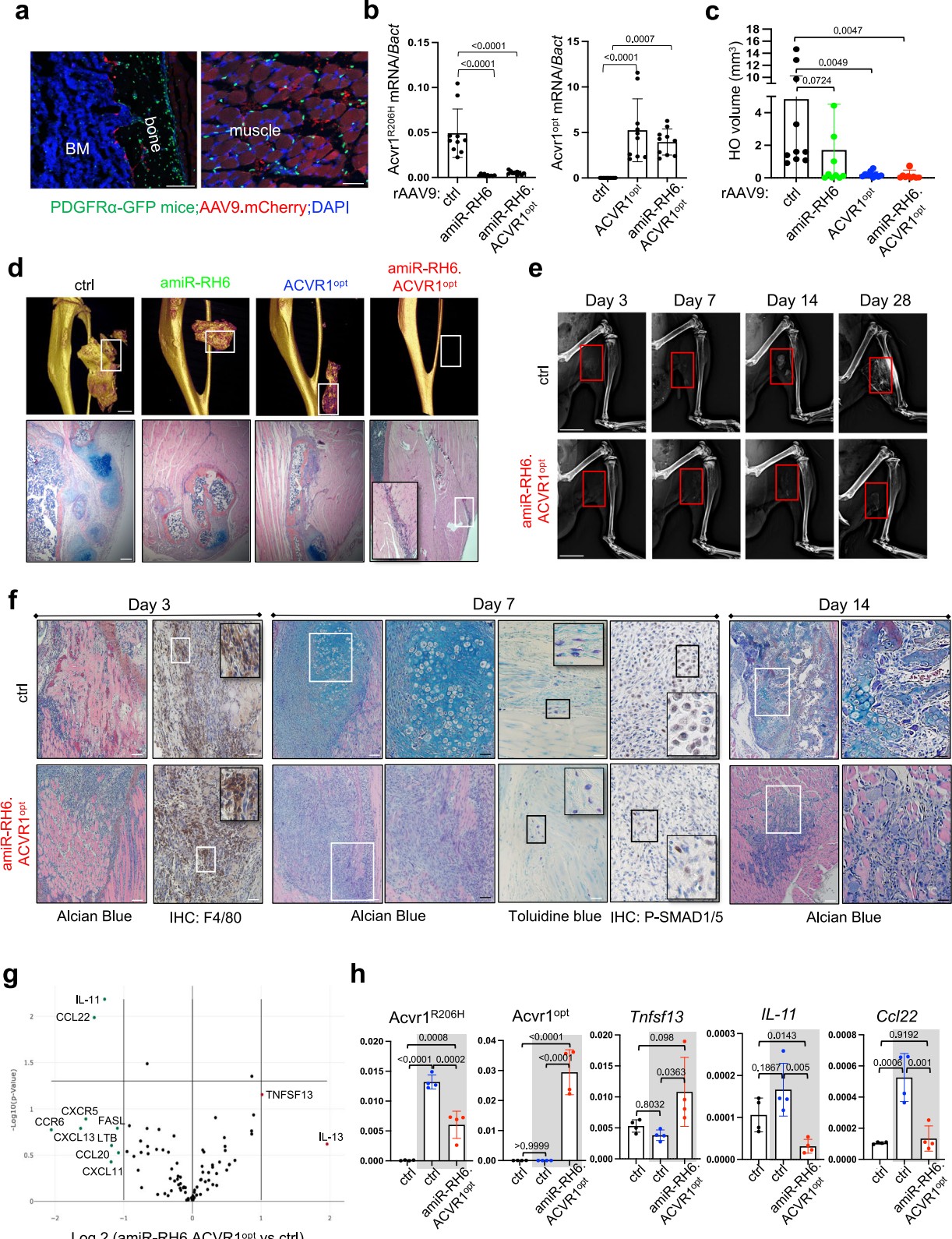

cervical spine, jaw, forelimb, hindlimb, hip, and ankle (Fig. 5d and h, Supplementary Fig. 11a). Remarkably, systemic delivery of *amiR-RH6.ACVR1opt* at birth almost completely prevented the incidence and severity of spontaneous HO throughout the body. Spontaneous HO was also reduced by treatment with *amiR-RH6* or *amiR-RH6.ACVR1opt* while *ACVR1opt* treatment showed a slight discrepancy between spontaneous HO incidence and severity (Fig. 5d, h, i). Daily i.p. injections of

2-week-old FOP mice with a retinoic acid receptor γ agonist (palovarotene) have been reported to cause untoward skeletal toxicities, including synovial joint overgrowth and growth plate deformity in the long bone[52]. As systemic delivery of rAAV9 at birth was not effective for the transduction of chondrocytes in the growth plates (Supplementary Fig. 8c), a single dose of i.v. injection with *amiR-RH6.ACVR1opt* at birth did not perturb the development of proliferating and hypertrophic

**Fig. 4 | Systemic delivery of AAV gene therapy at birth prevents traumatic HO in FOP mice. a** P1 PDGFRα-GFP reporter neonates ($n = 3$) were i.v. injected with $10^{11}$ GCs of rAAV9.mCherry and 2 weeks later, mCherry and GFP expression in AAV-treated tibia and muscle was assessed by fluorescence microscopy. BM bone marrow. Scale bars: 100 μm, **left**; 50 μm, **right**. **b**–**d** P1 *Acvr1$^{(R206H)Fl}$;Cre-ER$^{T2}$* neonates ($n = 10$) were i.v. injected with $10^{11}$ GCs of rAAV9 carrying *EGFP control, amiR-RH6, ACVR1$^{opt}$*, or *amiR-RH6.ACVR1$^{opt}$* and 6 weeks later, mice were treated with tamoxifen (10 mg/kg). Muscle injury was applied to the gastrocnemius muscle 3 days post-tamoxifen treatment. 4 weeks later, mRNA levels of *ACVR1$^{R206H}$* and *ACVR1$^{opt}$* were measured by RT-PCR (**b**, $n = 10$) and HO in the gastrocnemius muscle was assessed by microCT and histology (**c**, **d**). 3D reconstruction images (**d**) and quantification of HO volume (**c**, $n = 10$) are displayed. Alcian blue staining of HO tissues (**d**) was performed to assess chondrogenic anlagen. Scale bars: 1 mm, **top**; 200 μm, **bottom**. **e**–**h** To investigate the progression of HO pathogenesis, P1 *Acvr1R$^{(R206H)Fl}$* or *Acvr1R$^{(R206H)Fl}$;Cre-ER$^{T2}$* neonates ($n = 3$) were i.v. injected with $10^{11}$ GCs of rAAV9 carrying *EGFP control* or *amiR-RH6.ACVR1$^{opt}$* and 6 weeks later, mice were i.p. injected with tamoxifen. 3 days later, a Muscle injury was applied to the gastrocnemius muscle, and HO pathogenesis was assessed at a series of time points post-injury by radiography (heterotopic bone, **e**), at Day 3 by immunohistochemistry for F4/80 (monocytes/macrophages, **f**) and at Day 7 by Alcian blue staining (fibrosis, chondrogenesis, **f**) Toluidine blue staining (mast cells, **f**), and phospho-SMAD1/5 (BMP signaling, **f**). In (**e**), the red boxes indicate injured areas. RT$_2$ profiler PCR array (**g**) and RT-PCR analysis (**h**, $n = 4$) for inflammatory gene expression were performed on the gastrocnemius muscle 3 days post-injury (day 3). A scatter plot was generated from multiple t-test. AAV-treated *Acvr1R$^{(R206H)Fl}$* (control) and *Acvr1R$^{(R206H)Fl}$;Cre-ER$^{T2}$* muscle RNA with and without *amiR-RH6.ACVR1$^{opt}$* (gray boxes) are displayed (**h**). Scale bars: 5 mm, **e**; 100 μm, **f** Values represent mean ± SD by one-way ANOVA (**b**, **c**, **h**).

chondrocytes in the growth plate and articular cartilage (Fig. 5j, Supplementary Fig. 11a). Thus, unlike palovarotene that requires daily dosing and shows potential side effects, a single dose of systemic delivery of rAAV9.*amiR-RH6.ACVR1$^{opt}$* at birth is sufficient to prevent the early onset, spontaneous HO in *Acvr1$^{(R206H)KI}$* FOP mice without disturbing cartilage development.

Finally, to visualize how *amiR-RH6.ACVR1$^{opt}$* treatment prevents the pathogenesis of spontaneous HO in FOP mice, *Acvr1$^{(R206H)Fl}$;PDGFRα-cre* mice were further crossed with *PDGFRα-GFP* reporter mice and P1 *Acvr1$^{(R206H)Fl}$;PDGFRα-cre;PDGFRα-GFP* (*PDGFRα-GFP;Acvr1$^{R206H}$*) mice were i.v. injected with rAAV9 carrying mCherry control or *amiR-RH6.ACVR1$^{opt}$*. Five weeks later, *PDGFRα$^+$* FAPs in the skeletal muscle was monitored by fluorescence microscopy using GFP expression (Fig. 5k). As expected, a subset of GFP-expressing *PDGFRα$^+$* FAPs expressed mCherry, confirming AAV's transduction to *PDGFRα$^+$* FAPs within forming HO lesions. mCherry control-expressing *PDGFRα$^+$* FAPs differentiated into heterotopic bone-forming osteoblasts and primarily resided within forming HO lesions. By contrast, little to no evidence of heterotopic bone or chondrogenic anlagen in the skeletal muscle expressing *amiR-RH6.ACVR1$^{opt}$* (Supplementary Fig. 11b); GFP-expressing *PDGFRα$^+$* FAPs were primarily present in muscle interstitium (Fig. 5k), similar to WT *PDGFRα$^+$* FAPs (Fig. 4a, right). Thus, systemic delivery of *amiR-RH6.ACVR1$^{opt}$* at birth is likely to suppress the initiation process of spontaneous HO in the skeletal muscle. Our data demonstrate that with a single dose of i.v. injection at birth, rAAV9.*amiR-RH6.ACVR1$^{opt}$* almost completely prevents spontaneous HO during early skeletogenesis and trauma-induced HO at early adulthood. Overall mobility, activity levels, and body stature were also substantially ameliorated in these mice (Supplementary Movie 1).

## Early adulthood treatment with AAV gene therapy prevents spontaneous HO in adult *Acvr1$^{(R206H)KI}$* FOP mice

To examine tissue biodistribution of systemically delivered rAAV9 during early adulthood in *Acvr1$^{(R206H)KI}$* FOP mice, 6-week-old *Acvr1$^{(R206H)Fl}$;Cre-ER$^{T2}$* mice were i.v. injected with rAAV9.*mCherry* three days after tamoxifen treatment. 12 weeks later, mCherry expression in the heart, kidney, liver, and skeletal muscle and knockdown efficiency of *ACVR1$^{R206H}$* and expression of *ACVR1$^{opt}$* in the liver were validated. However, mouse *Acvr1* expression was unchanged in these mice (Fig. 6a, Supplementary Fig. 12a–c). Alternatively, these mice were i.v. injected with a single dose of rAAV9.*amiR-RH6.ACVR1$^{opt}$* and 12 weeks later, control-treated *Acvr1$^{(R206H)Fl}$;Cre-ER$^{T2}$* mice progressively developed osteochondromas in the tibia, osteoarthritis in the knees, and spontaneous HO at multiple anatomical locations, including the cervical spine, hips, and knees−skeletal features that are commonly seen in individuals with FOP[53]. Systemic delivery of *amiR-RH6.ACVR1$^{opt}$* at early adulthood prevented spontaneous HO from the cervical spine (Fig. 6b and c), while total HO mass and incidence throughout the body were substantially reduced in

these mice (Fig. 6d–f). Additionally, control-treated mice often developed HO bridging the femur to the fibular head (Fig. 6g, Supplementary Movies 2 and 3) and severe osteoarthritis in the knees (Fig. 6h), resulting in the immobility of hindlimbs in *Acvr1$^{(R206H)KI}$* FOP mice. Thus, *Acvr1$^{(R206H)Fl}$;Cre-ER$^{T2}$* mice resemble multiple clinical FOP features found in human FOP patients, including progressive HO, an orthotopic fusion of cervical vertebrae, a fusion of thoracic and lumbar vertebrae, osteochondromas, and early onset degenerative joint disease[53], which were all substantially prevented by the treatment with *amiR-RH6.ACVR1$^{opt}$* (Fig. 6i). Accordingly, these mice displayed normal mobility, activity levels, and body posture (Supplementary Movie 4). Moreover, tissue morphology and structure in *amiR-RH6.ACVR1$^{opt}$*-treated mice are largely normal, suggesting little to no obvious anatomic off-target side effects of rAAV9.*amiR-RH6.ACVR1$^{opt}$* in non-HO tissues of FOP mice (Supplementary Fig. 12d). Previous studies demonstrated that BMP signaling plays critical roles in skeletal growth, joint patterning, and cartilage development during skeletogenesis, and that dysregulated BMP signaling by the *ACVR1$^{R206H}$* mutation disturbs these procedures[41,54]. However, tamoxifen treatment of *Acvr1$^{(R206H)Fl}$;Cre-ER$^{T2}$* mice in early adulthood did not develop any apparent abnormalities in growth plates in the presence of control or *amiR-RH6.ACVR1$^{opt}$* (Fig. 6h), suggesting minimal effects of the *ACVR1$^{R206H}$* mutation on the development of growth plates during early adulthood.

FOP patients have chronically elevated levels of hyperinflammatory immune cells and pro-inflammatory cytokines[35,55,56]. FOP macrophages and mast cells are primed toward inflammatory responses[57], and depletion of these cells reduces HO in mice[44]. To examine the potential impact of i.v.-injected rAAV9.*amiR-RH6.ACVR1$^{opt}$* on systemic immunity in *Acvr1$^{(R206H)KI}$* FOP mice, complete blood counts, flow cytometry for immune cells, and spleen histology were performed in 18-week-old *Acvr1$^{(R206H)Fl}$* (control) and *Acvr1$^{(R206H)Fl}$;Cre-ER$^{T2}$* mice treated with control or *amiR-RH6.ACVR1$^{opt}$*. The numbers of platelets and immune cells, including T cells, B cells, dendritic cells, monocytes, macrophages, and neutrophils in the plasma and spleen, were comparable between control- and *amiR-RH6.ACVR1$^{opt}$*-treated mice (Fig. 6j, Supplementary Fig. 13a and b). Likewise, there was little to no alteration of germinal center architecture in AAV-treated spleens (Supplementary Fig. 13c). These results suggest that a single dose of i.v. administered rAAV9.*amiR-RH6.ACVR1$^{opt}$* is a potent inhibitor of spontaneous HO in adult FOP mice but has no grossly apparent effects on cellular or tissue components of the immune system. Taken together, an AAV-mediated combination gene therapy that executes *ACVR1$^{R206H}$* allele-specific silencing and *ACVR1$^{opt}$* expression at birth or early adulthood is a promising approach to prevent disabling HO, providing the potential for clinical translation to FOP patients.

## Discussion
More than 95% of FOP patients have a heterozygous, ACVR1[R206H]-activating mutation that promotes dysregulated BMP pathway signaling

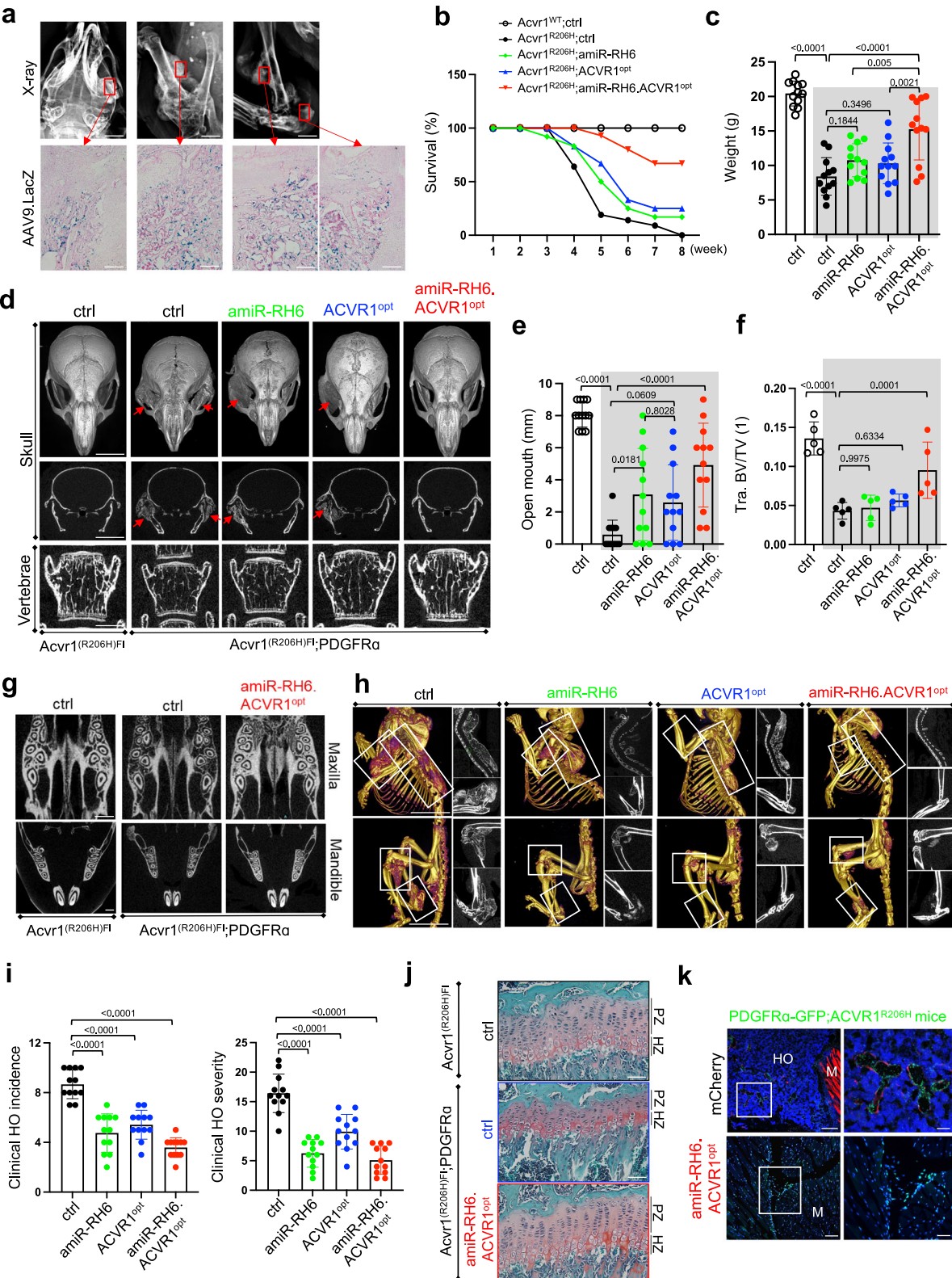

and subsequent transformation of soft connective tissues to heterotopic bone. AAV gene therapy is attractive for treating patients with FOP, due to the highly recurrent genetic cause, the lifelong progression of the severe extraskeletal formation, and the high burden of taking a lifelong medication. In this study, we developed three AAV-based gene therapy approaches for FOP, including dilution of ACVR1$^{R206H}$ receptor by over-expression of WT ACVR1 receptor (gene replacement), *ACVR1$^{R206H}$* allele-specific silencing by an AAV-compatible artificial miRNA, and the combination of gene replacement and silencing. These approaches function as potent suppressors of Activin A-induced aberrant BMP signaling, chondrogenesis, and osteogenesis of human FOP iPSCs and mouse *ACVR1$^{(R206H)K1}$* skeletal cells while selectively inhibiting and/or diluting aberrant BMP pathway signaling by the *ACVR1$^{R206H}$* mutation. We have shown that both t.d. and i.v. administration of the rAAV9 serotype can target early- and late-stages of HO-inducing cells, including FAP-lineage cells and mature

**Fig. 5 | Systemic delivery of AAV gene therapy at birth prevents spontaneous HO in juvenile FOP mice. a** $5 \times 10^{13}$ vg/kg of rAAV9.*LacZ* was i.v. injected into 3-week-old male or female *Acvr1*$^{(R206H)Fl}$;*PDGFRα-cre* mice (*n* = 3) and, 2 weeks later, radiography of the whole body was performed to locate HO lesions. HO tissues were stained for β-galactosidase. Scale bars: 2 mm, **top**; 100 μm, **bottom**. For data shown in **b–j**, P1 *Acvr1*$^{(R206H)Fl}$ or *Acvr1*$^{(R206H)Fl}$;*PDGFRα-cre* neonates (*n* = 12) were i.v. injected with $10^{11}$ GCs of rAAV9 and AAV-treated mice were analyzed weekly up to 8 weeks of age. **b** and **c** Survival curve (**b**) and body weight (**c**) for the AAV-treated groups. **d** MicroCT analysis for skulls and lumbar vertebrae (L4) of 4 to 5-week-old mice are shown in 3D reconstructed images (**d, top**) and 2D transverse sections (**d, middle and bottom**). Scale bars: 5 mm, **top, middle**; 1 mm, **bottom**. Arrows indicate temporomandibular joint ankylosis. **e** and **f** Plots showing the distance of open mouth in AAV-treated mice (**e**, *n* = 12) and the quantification of vertebral bone mass (**f**, *n* = 5). AAV-treated *Acvr1R*$^{(R206H)Fl}$ and *Acvr1*$^{(R206H)Fl}$;*PDGFRα-cre* mice (gray boxes) are displayed (**c, e, f**). Tra. BV/TV trabecular bone volume per tissue volume. **g** MicroCT analysis showing the maxillary and mandibular bone mass. Scale bar: 1 mm. **h** MicroCT analysis showing spontaneous HO from whole body scans of AAV-treated mice. 3D reconstructed images (**left**) and 2D transverse sections (**right**) are displayed. Scale bars: 4 mm. **i** Clinical HO incidence and severity was scored using whole-body microCT and radiography (*n* = 12). **j** Safranin O staining of AAV-treated tibias showing normal chondrocyte zones in the growth plate. Scale bars: 50 μm. **k** P1 *Acvr1*$^{(R206H)Fl}$;*PDGFRα-cre*;*PDGFRα-GFP* reporter neonates were i.v. injected with $10^{11}$ GCs of rAAV9 (*n* = 3) and 5 weeks later, mCherry- and/or GFP-expressing cells were visualized by fluorescence microscopy. The right panels are enlarged images of the white-boxed regions on the left. Scale bars: 100 μm, **left**; 50 μm, **right**. Values represent mean ± SD by one-way ANOVA test (**c, e, f, i**).

osteoblasts. Importantly, AAV9-mediated gene therapy is effective when introduced at birth and early adulthood and can suppress both traumatic and spontaneous HO without causing detrimental effects on cartilage development, bone growth, or bone remodeling. Thus, our findings provide the first in vivo evidence that AAV-based gene therapy is a promising option for the prevention of HO in FOP. The feasibility and obstacles of AAV-based gene therapy for FOP were further discussed in a recent review article[58].

In contrast to commercialized gene therapy for lipoprotein lipase deficiency[59], inherited retinal dystrophy[60], and spinal muscular atrophy[61], which introduces genes encoding missing proteins or encoding corrective proteins, our gene therapy approaches for FOP were designed to (1) dilute dysregulated BMP signaling effect of the mutant ACVR1$^{R206H}$ receptor with the WT ACVR1 receptor, (2) silence the expression of the ACVR1$^{R206H}$ receptor at the mRNA level using *ACVR1*$^{R206H}$ allele-specific amiR, or (3) remove the effects of the ACVR1$^{R206H}$ receptor and express WT ACVR1 receptors simultaneously (Supplementary Fig. 1). While the therapeutic effects of these three approaches on trauma-induced HO were comparable in FOP mice with transdermal administration when systemically delivered at birth, the combination gene therapy was most effective in suppressing both traumatic and spontaneous HO. This discrepancy may result from lower transduction efficiency of rAAV9 to targeted tissues via systemic delivery than direct local delivery. Alternatively, differential expression levels of ACVR1$^{R206H}$ vs. ACVR1$^{WT}$ within forming HO lesions at different anatomical locations may affect the therapeutic efficacy of these three different approaches on spontaneous HO in *ACVR1*$^{(R206H)KI}$ FOP mice. Nonetheless, further studies will be necessary to define the stoichiometry of ACVR1$^{R206H}$ and WT ACVR1 receptors in AAV-treated HO lesions and the long-term therapeutic outcomes of AAV gene therapy at different doses in preventing both traumatic and spontaneous HO in *ACVR1*$^{(R206H)KI}$ FOP mice. Finally, correction of the classic *ACVR1*$^{R206H}$ mutation (c.617G>A) at the genomic level using the clustered regularly interspaced short palindromic repeats (CRISPR)/Cas9-based adenine base editor (ABE) system was considered for FOP treatment by directly converting adenine to guanine in the human *ACVR1*$^{R206H}$ allele without creating double-stranded DNA breaks[62]. However, it is challenging to use an AAV-based CRISPR/ABE system as in vivo gene therapy for FOP because ABE is a large bacteria-derived protein that exceeds the AAV packaging size limit (~4.7 kb), and this system has low gene editing efficiency and high immunogenicity.

Our data demonstrate for the first time that the rAAV9 serotype is a highly effective vector for transducing FAP-lineage cells and osteoblasts within forming HO lesions. Accordingly, rAAV9-mediated gene therapy suppressed HO in *ACVR1*$^{(R206H)KI}$ FOP mice when administered t.d. at early adulthood or i.v. at birth. Notably, despite a high transduction efficiency of the rAAV6.2 serotype in human and mouse osteogenic cells in vitro, rAAV6.2 was not effective for in vivo transduction of FAPs and osteoblasts in forming HO lesions. This discrepancy may be due to multiple physiological barriers, including the route of administration, serum factors, circulating neutralizing antibodies, and trans-vascularity[25]. rAAV9-mediated gene therapy is currently the leading platform for treating neurological and musculoskeletal disorders, such as Parkinson's disease[63], amyotrophic lateral sclerosis (ALS)[64], type 1 spinal muscular atrophy (SMA)[65], and Duchenne muscular dystrophy (DMD)[66], in part, due to rAAV9's ability to target the central nervous system (CNS) and skeletal muscle when i.v. administered[67,68]. Previous studies have demonstrated that activating mutations in the human ACVR1 receptor that cause FOP, including the *ACVR1*$^{R206H}$ mutation (c.617G>A)[4], are also involved in the tumorigenesis of diffuse intrinsic pontine glioma (DIPG)[69,70]. However, the activating mutations of ACVR1 in FOP are germline-driving mutations whereas, in DIPG, they are somatic within the tumor and are not associated with HO. DIPG is a pediatric brain tumor with a highly infiltrative malignant glial neoplasm of the ventral pons. The median survival time is 9–12 months, but neither surgical resection nor chemotherapeutic agents show any substantial survival benefit in clinical trials. Since treatment with ACVR1 inhibitors markedly prolonged the survival of DIPG mice[71–73], brain-transducible AAV9-mediated gene therapy for FOP might be also useful to treat tumorigenesis of DIPG. Further studies are needed to define the transduction efficiency of rAAV9 in the ventral pons and the therapeutic effects of rAAV9 vectors in DIPG mice with the *ACVR1*$^{R206H}$ mutation. Since this study mainly focused on HO in *ACVR1*$^{(R206H)KI}$ FOP mice, the effects of AAV gene therapy on non-HO phenotypes in FOP may not be addressed.

Recently, a rAAV9 vector carrying *SMN1* (rAAV9.SMN1, Zolgensma) was FDA-approved as gene therapy to treat spinal muscular atrophy. Systemic infusion of a single high dose of Zolgensma ($2 \times 10^{14}$ vg/kg) has been proven therapeutically effective and safe in clinical trials. However, the high transduction efficacy of systemically delivered rAAV9 vectors to the diseased liver can cause untoward side effects, including acute serious liver injury and liver failure[74]. Similar to Zolgensma, our gene therapy for FOP utilizes a systemic infusion of rAAV9 capsid to deliver *amiR-RH6.ACVR1*$^{opt}$, but at a much lower dose for targeting skeletal muscle and bone cells instead of the motor neuron. As tested in mouse models of FOP, our AAV gene therapy showed robust therapeutic effects at $2–5 \times 10^{13}$ vg/kg, which is 4–10 fold lower than an FDA-approved dose of Zolgensma ($2 \times 10^{14}$ vg/kg). However, to minimize potential side effects by systemically delivered rAAV9 vectors, further vector improvements using endogenous miRNA-mediated liver-detargeting and/or capsid modification will be necessary to limit *amiR-RH6.ACVR1*$^{opt}$ expression in the liver and/or to specifically deliver *amiR-RH6.ACVR1*$^{opt}$ to HO-causing cells, respectively. Alternatively, since a single dose ($2–5 \times 10^{12}$ vg/kg) of rAAV9.*amiR-RH6.ACVR1*$^{opt}$ via transdermal injection to the skeletal muscle was highly effective in suppressing trauma-induced HO (Fig. 3i and j), we anticipate that direct transdermal injection of the AAV vectors to the flare-up lesions in the skeletal muscle/skeleton can be effective at doses 10 or 100 fold lower than the dose used for IV injection of rAAV9.*amiR-RH6.ACVR1*$^{opt}$ or Zolgensma, respectively.

There are several limitations to testing the therapeutic efficacy of the AAV vectors using FOP mouse models of *Acvr1*$^{(R206H)Fl}$: (1) Adult

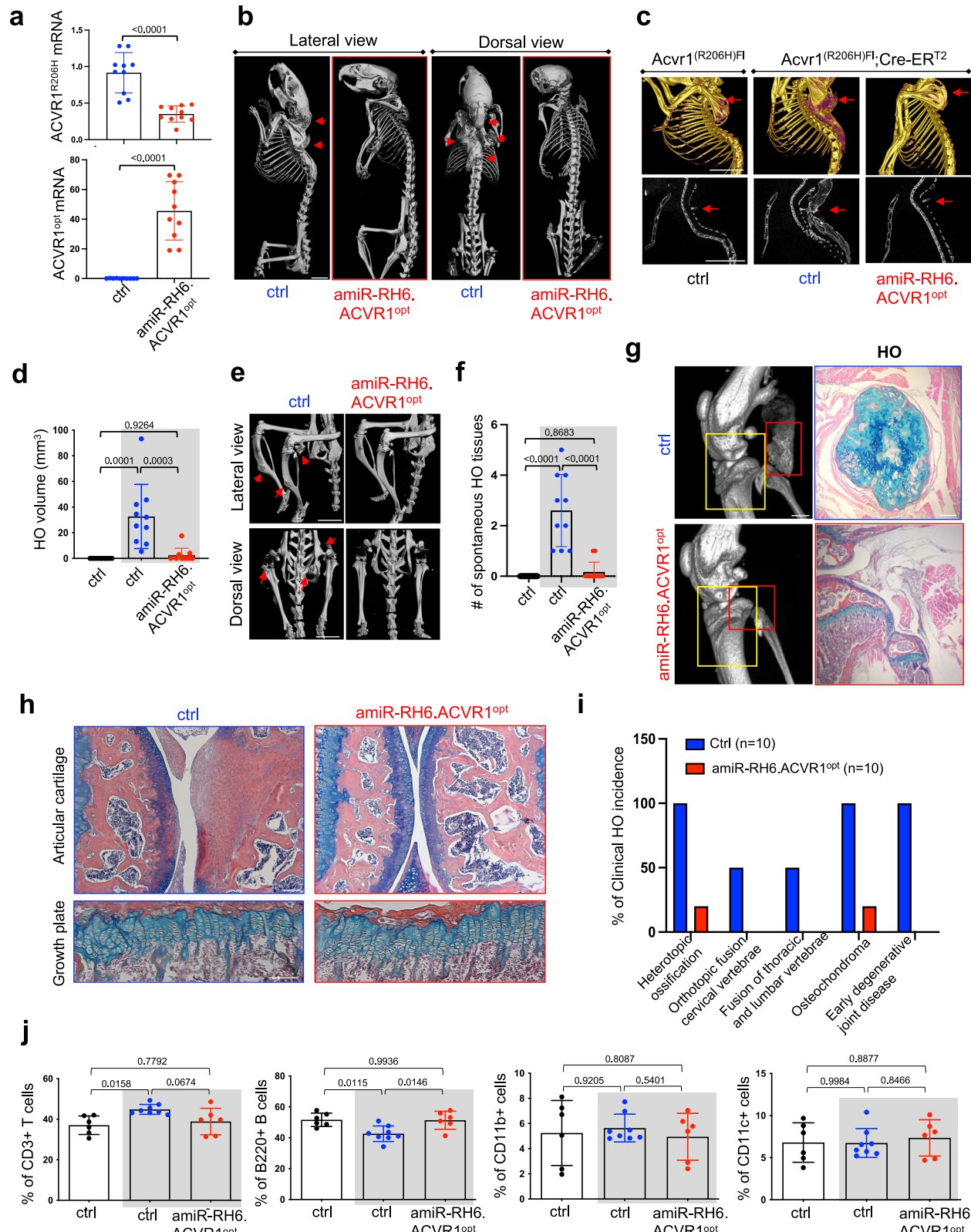

*Acvr1*[(R206H)Fl]*;*Cre-ER[T2] mice with tamoxifen-induced expression of *ACVR1*[R206H] receptor after normal skeletal development may not recapitulate developmental aspects of FOP contributing to later HO phenotypes. (2) Since *Acvr1*[(R206H)Fl]*;*PDGFRα-cre and tamoxifen-induced *Acvr1*[(R206H)Fl]*;*Cre-ER[T2] mice express *ACVR1*[R206H] receptor only in a subset of HO-causing cells, these mouse models may not recapitulate the full spectrum of FOP phenotypes seen in patients with FOP. (3) While AAV

treatment appears effective in preventing both spontaneous and traumatic HO in FOP mice, the potential for systemic toxicities or effects on non-skeletal cells remains unexplored. Thus, we would speculate that the initial use of AAV gene therapy might be most applicable for FOP patients with a localized early-stage flare-up or as a preventative for subjects undergoing surgeries. However, future investigation for vector biodistribution, toxicity, and dose-ranging in

**Fig. 6 | Systemic delivery of AAV gene therapy in early adulthood prevents spontaneous HO in adult FOP mice.** $5 \times 10^{13}$ vg/kg of rAAV9 carrying EGFP control or *amiR-RH6.ACVR1$^{opt}$* was i.v. injected into 6-week-old female *Acvr1$^{(R206H)Fl}$* or *Acvr1$^{(R206H)Fl}$;Cre-ER$^{T2}$* mice (*n* = 10) 3 days after tamoxifen treatment. 12 weeks later, mRNA levels of *ACVR1$^{R206H}$* and *ACVR1$^{opt}$* in the liver were assessed by RT-PCR (**a**) MicroCT analysis showing whole body (**b**), torso (**c**), and lower body (**e**) of AAV-treated mice. Arrows indicate HO lesions. Scale bars: 5 mm, **b**; 1 mm, **c**, **e** Total HO volume (**d**, *n* = 10) and numbers of HO lesions (**f**, *n* = 10) throughout the body were quantitated. MicroCT (**g**, **left**) and histology of knee joints were performed to assess bridging HO (**g**, **right**), degeneration of articular cartilage (**h**, **top**), and chondrocytes in the growth plate (**h**, **bottom**). In **g**, the red box, bridging HO; yellow box, articular cartilage, and growth plate. Scale bars: 1 mm, **g**, **left**; 100 μm, **g**, **right**. Total percentage of clinical HO incidence in AAV-treated mice was assessed (**i**) The frequency of immune cells within the population of total splenocytes suggests that rAAV9.*amiR-RH6.ACVR1$^{opt}$* has little to no effect on systemic immunity (**j**, *n* = 6–8). AAV-treated *Acvr1R$^{(R206H)Fl}$* (control) and *Acvr1$^{(R206H)Fl}$;Cre-ER$^{T2}$* mice (gray boxes) are displayed (**d**, **f**, **j**). Values represent mean ± SD by an unpaired two-tailed Student's *t*-test (**a**) or ANOVA test (**d**, **f**, **j**).

large animals is required before any consideration can be given to applying AAV gene therapy to individuals who have FOP.

Our proof-of-concept studies demonstrate AAV gene therapy is a potent inhibitor of traumatic and spontaneous HO in FOP mice, providing a potential clinical translation to FOP patients. However, since immunological triggers are known to pose a high risk for HO induction in FOP patients, any consideration of AAV gene therapy needs to be scrupulously approached. Although rAAVs have low post-infection immunogenicity[15], and FOP mice showed little to no immune responses to systemically delivered AAVs, a high dose administration of rAAVs could potentially induce systemic immune responses in FOP patients complicating reliable delivery of therapeutic genes to HO-causing cells and may compromise the subsequent safety of this method as well as any therapeutic benefit. A cocktail therapy of rAAVs with immunosuppressors, FOP inhibitors, and/or non-immunogenic liposomes or nanoparticles might minimize FOP-associated immune responses. Alternatively, AAV capsid-specific chimeric antigen receptor-expressing regulatory T cells (AAV-CAR Tregs) have been shown to suppress immune responses to AAV capsids and transgenes[75] and might be engineered for AAV gene therapy for FOP. Finally, the long-term durability and safety of therapeutic gene expression are of paramount importance in considering these approaches for potential use in FOP patients where lifelong HO suppression will be necessary.

## Methods

This study complies with all relevant ethical regulations including the University of California San Francisco Institutional Review Board (IRB) and the University of Massachusetts Chan Medical School Institutional Animal Care and Use Committee (IACUC) and Institutional Biosafety Committee (IBC).

### Plasmids

To screen human ACVR1$^{R206H}$-specific amiRs, endogenous complementary DNA sequences of human *ACVR1$^{R206H}$*, *ACVR1$^{WT}$*, or *ACVR1$^{opt}$* were inserted into 3-UTR of the Renilla luciferase gene of pcDNA3 RLUC POLIRES FLUC (Addgene, #45642, Supplementary Fig. 2a). Mammalian expression vectors (pcDNA6) encoding human *ACVR1$^{WT}$* and *ACVR1$^{R206H}$* ORF were obtained from Dr. Hyun-Mo Ryoo (Seoul National University, Supplementary Fig. 2b)[76].

### rAAV vector design and production

The pAAVsc-*CB6*-ACVR1$^{opt}$ was generated by replacing the *mCherry* reporter with a codon-optimized version of the human ACVR1 complementary DNA (*ACVR1$^{opt}$*), and then, the chicken β actin (CBA) intron in the plasmid was replaced with the MassBiologics (MBL) or synthetic intron to reduce the AAV vector genome size (Fig. 1a). The artificial miRNA (amiR) against human *ACVR1$^{R206H}$* was designed by using a custom Excel macro, which considers miR-33 scaffold design rules to generate optimized amiR cassettes. The tool will be shared upon request. Plasmids were constructed by Gibson assembly and standard molecular biology methods. DNA sequences for *amiR-33-ctrl* and *amiR-33-human ACVR1$^{R206H}$* were synthesized as gBlocks and cloned into the intronic region of the pAAVsc-*CB6-mCherry* plasmid at the restriction enzyme sites (PstI and BglII, Fig. 1c)[77]. Constructs were verified by sequencing. Additionally, the pAAVsc-CB6-*Egfp* construct was packaged into AAV1 (1.8E + 13 GC/ml), AAV2 (1.5E + 12 GC/ml), AAV2TM (1.5E + 12 GC/ml), AAV3 (6E + 12 GC/ml), AAV4 (6.5E + 12 GC/ml), AAV5 (2.4E + 13 GC/ml), AAV6 (8E + 12 GC/ml), AAV6.2 (8E + 12 GC/ml), AAV7 (1.5E + 13 GC/ml), AAV8 (7E + 12 GC/ml), AAV9 (1.5E + 13 GC/ml), AAVrh8 (8E + 12 GC/ml), AAVrh10 (8E + 12 GC/ml), AAVrh39 (1.0E + 13 GC/ml), and AAVrh43 (6E + 12 GC/ml) capsids. Alternatively, the constructs of pAAVsc-CB6.*mCherry* or pAAVsc-CB6.*LacZ* were packaged into AAV6.2 (1.0E + 13 GC/ml) or AAV9 (1.0E + 13 GC/ml) capsids. rAAV production was performed by transient transfection of HEK293 cells, purified by CsCl sedimentation, and titered by droplet digital PCR (ddPCR) on a QX200 ddPCR system (Bio-Rad) using the *Egfp or mCherry* prime/probe set[78]. The sequences of gBlocks™ and oligonucleotides for ddPCR are listed in Supplementary Table 1.

### Cell culture and reagents

HEK293T cells (#CRL-3216) and C2C12 cells (#CRL-1772) were purchased from ATCC and human bone marrow-derived mesenchymal stromal cells (BMSCs, #7500) were purchased from ScienCell Research Laboratories. They were cultured according to the manufacturer's manuals. Human adipose tissue-derived stromal cells (ASC) were kindly gifted from Dr. Silvia Corvera (UMass Chan Medical School) and cultured as described in the previous publications[28,29]. Induced pluripotent stem cells (iPSC) were generated from dermal fibroblasts of healthy human donors (WT) or FOP patients by Dr. Edward Hsiao (UCSF) and cultured as described in the previous publications[26,27]. Mouse fibroadipogenic progenitors (FAPs) were FACS sorted from the digested skeletal muscle of 4-week-old *Acvr1$^{(R206H)Fl}$;PDGFRα-cre* mice using cell surface markers (PDGFRα$^{+}$Sca1$^{+}$CD31$^{-}$CD45$^{-}$) and cultured in DMEM (Corning) containing 20% FBS (Corning), and 1% penicillin/streptomycin (Corning)[11]. Mouse bone marrow-derived stromal cells (BMSCs) were isolated from 4-week-old *Acvr1$^{(R206H)Fl}$* or *Acvr1$^{(R206H)Fl}$;PRRX1-cre* femurs. Cells were maintained in α-MEM medium (Gibco) containing 10% FBS (Corning), 2 mM ʟ-glutamine (Corning), 1% penicillin/streptomycin (Corning), and 1% non-essential amino acids (Corning) while they were differentiated into mature osteoblasts under osteogenic medium containing ascorbic acid (200 μM, Sigma, #A8960) and β-glycerophosphate (10 mM, Sigma, #G9422). Mouse chondrogenic precursors were isolated from the knee joints of P2 *Acvr1$^{(R206H)Fl}$;PRRX1-cre* neonates using collagenase D (Sigma, #11088866001) and cultured in DMEM (Corning) containing 10% FBS (Corning), 2 mM ʟ-glutamine (Corning), and 1% penicillin/streptomycin (Corning)[79]. Recombinant BMP4 (#314-BP), BMP2/7 (#3229-BM), and Activin A (#338-AC) proteins were purchased from R&D systems.

### Human iPSC culture

The generation of human iPSCs was previously described[27]. Briefly, de-identified human dermal fibroblasts from commercial sources or from 3-mm skin biopsies carefully obtained from donors (without causing deep tissue injury) or from surgical excess were cultured. Cells less than five passages old were used for iPS cell reprogramming. The presence or absence of the ACVR1 mutation was sequenced and verified as described[4]. Retroviral[80] and episomal integration-free iPS cells[81]

were derived as described. iPSC lines were maintained in a primate ES cell medium (ReproCELL, Tokyo, Japan) on irradiated SNL feeder cells. SNLs were carefully removed by at least one passage in feeder-free conditions before use in differentiation assays. All human tissue collection, human stem cell studies, procedures, and written consents were approved by the UCSF Committee on Human Research, the UCSF Gamete and Embryonic Stem Cell Research Committee, or by the Ethics Committee of the Department of Medicine.

## Mice

Mice were housed in standard cages in a temperature-controlled room (22–24 °C) with a 12 h dark–light cycle and fed with standard chow (LabDiet, #5P7622;.22.5% protein, 5.4% fat, 4% fiber, 50% polysaccharide). Mice harboring a conditional *Acvr1*[R206H] knock-in allele (*Acvr1*[(R206H)Fl])[82] were kindly gifted from the International FOP Association via Dr. Daniel Perrien (Emory University) and maintained on C57BL/6J background. The target constructs described in Supplementary Fig. 7c were inserted into the locus of mouse *Acvr1*. Wildtype mouse exons 5–10, neomycin-resistant gene (neo cassette), one F3 site, and one roxP site are deleted by Cre recombinase, resulting in the expression of human cDNA exons 6–11 harboring the R206H mutation and eGFP. *Acvr1*[(R206H)Fl] mice were crossed with Cre-ER[T2] mice (*Acvr1*[(R206H)Fl];Cre-ER[T2]) where tamoxifen-induced expression of Cre recombinase mediates *Acvr1*[R206H]-driven HO; Pdgfrα-Cre mice (*Acvr1*[(R206H)Fl];*Pdgfrα-cre*) where expression of Cre recombinase in PDGFRα[+] FAPs mediates *Acvr1*[R206H]-driven HO; Prrx1-Cre mice (*Acvr1*[(R206H)Fl];*Prrx1-cre*) where expression of Cre recombinase in Prrx1[+] skeletal progenitors in the limb mesenchyme mediates *Acvr1*[R206H]-driven HO. Cre-ER[T2], Pdgfrα-Cre, and Prrx1-Cre mice were purchased from Jackson Laboratory and maintained on C57BL/6J background. To label Pdgfrα-expressing FAPs, *Acvr1R*[(R206H)Fl];*Pdgfrα-cre* mice were further crossed with *Pdgfrα-GFP* reporter mice (Jackson Laboratory, C57BL/6J). To label Tie2-expressing FAPs, *Tie2-cre* mice (Jackson Laboratory, C57BL/6J) were crossed with Ai9-mCherry mice (Rosa26[mCherry], Jackson Laboratory). For postnatal activation of Cre-ER[T2], 100 mg/kg tamoxifen (Sigma, #T5648) in sunflower seed oil (Sigma, #S5007) was intraperitoneally (i.p.) injected into 6-week-old mice once a day for 5 consecutive days. Mouse genotypes were determined by PCR using tail genomic DNA. No sex-specific differences in HO phenotypes were observed in these mice. Primer sequences are available upon request. Control littermates were used and analyzed in all experiments. All animals were used in accordance with the NIH Guide for the Care and Use of Laboratory Animals and were handled according to protocols approved by the University of Massachusetts Chan Medical School Institutional Animal Care and Use Committee (IACUC).

## MicroCT and radiography

MicroCT (uCT35; SCANCO Medical AG; Bruttisellen, Switzerland) was used for qualitative and quantitative assessment of trabecular and cortical bone microarchitecture and performed by an investigator blinded to the genotypes of the animals under analysis. MicroCT scanning was performed at 55 kVp and 114 mA energy intensity with 300-ms integration time. Specific voxel size used for femur, maxilla and mandibular body is 7 and 12 μm for vertebrae. All images were reconstructed using image matrices of 1024 × 1024 pixels. For trabecular bone analysis of the distal femur, an upper 2.1 mm region beginning 280 μm proximal to the growth plate was contoured. For the cortical bone analysis of femurs, a midshaft region of 0.6 mm in length was used. L4 spinal segments were used for vertebrae analysis. 3D reconstruction images were obtained from contoured 2D images by methods based on distance transformation of the binarized images. Alternatively, the Inveon multimodality 3D visualization program was used to generate fused 3D viewing of multiple static or dynamic volumes of microCT modalities (Siemens Medical Solutions USA, Inc.). All images presented are representative of the respective genotypes ($n > 5$).

Trident Specimen Radiography system (Hologic, USA) was used to generate detailed radiographic images of the whole mouse body after euthanasia. The X-ray beam intensity was 1 mA 28–30 kV with AEC (automatic exposure control) for fast image acquisition.

## Histology and immunohistochemistry

For histological analysis, femurs, vertebrae, and skulls were dissected from AAV-treated mice, fixed in 10% neutral buffered formalin for 2 days at room temperature, and decalcified by 14% EDTA tetrasodium salt, pH 7.6 for 3–4 weeks at 4 °C. Samples were kept in 70% ethanol until processed on a vacuum infiltration tissue processor. Sections were done on a microtome (HistoCore Multicut; Leica, USA) at a thickness of 6 μm along the coronal plate from anterior to posterior. Slides were stained with hematoxylin and eosin (H&E), alcian blue hematoxylin-orange G, or toluidine blue.

For immunohistochemistry, paraffin sections were dewaxed and stained following the manufacturer's procedure using the Discovery XT automated immunohistochemistry stainer (Ventana Medical Systems, Inc., Tucson, AZ, USA). Citrate-based antigen unmasking solution (Vector Laboratories, H-3300) and BLOXALL endogenous blocking solution (Vector laboratories, SP-6000) were used for antigen retrieval and blocking, respectively. Sections were incubated with antibodies specific to F4/80 (1:100, Cell Signaling Technology, #70076), phospho-SMAD1/5 (1:100, Cell Signaling Technology, #9516) overnight at 4 °C, and a secondary antibody of VisUCyte[TM] HRP Polymer antibodies for 40 min at room temperature, then incubated with the substrate working solution (DAB substrate kit, Vector Sk-4100) at room temperature for 1–2 min followed by hematoxylin staining (Vector laboratories H-3404). Reaction buffer (pH 7.6 Tris buffer) was used as a washing solution. The stained samples were visualized either using an Olympus microscope Bx50 (Olympus, USA) or EVOS M7000 (Life Technology, USA).

For frozen sectioning, dissected specimens were fixed in 4% paraformaldehyde for 2 –3 days followed by 15 days of semi-decalcification using 14% EDTA tetrasodium salt, pH 7.6 at 4 °C. Infiltration was processed using 20% sucrose solution prior to OCT embedding. Slides were prepared on Cryostat (LM3050s; Leica, USA) at a thickness of 12 μm.

## Quantitative RT-PCR, RT² profiler PCR arrays, and immunoblotting

The total RNA was purified from cells and tissues using QIAzol (QIAGEN) and cDNA was synthesized using the High-Capacity cDNA Reverse Transcription Kit (Applied Biosystems, #4368814). Quantitative RT-PCR was performed using iTaq[TM] Universal SYBR® Green Supermix (Bio-Rad, #1725122) with CFX connect RT-PCR detection system (Bio-Rad). To measure mRNA levels of the indicated genes in the injured areas or HO lesions of AAV-treated mice, the tibialis muscles were snap-frozen in liquid nitrogen for 30 s and in turn homogenized in 1 ml of QIAzol for 1 min. Alternatively, HEK293T cells, human WT or FOP iPSCs, mouse FAPs, and osteogenic or chondrogenic progenitors were lysed using QIAzol and total RNA was subjected to RT-PCR analysis. Primers used for PCR are described in Supplementary Table 1. Finally, RT² profiler PCR arrays (QIAGEN) were used to measure mRNA levels of 84 inflammatory cytokines and receptors, including *Ccl1, Ccl11, Ccl12, Ccl17, Ccl19, Ccl2,*

*Ccl20, Ccl22, Ccl24, Ccl3, Ccl4, Ccl5, Ccl6, Ccl7, Ccl8, Ccl9, Cx3cl1, Cxcl1, Cxcl10, Cxcl11, Cxcl12,Cxcl13, Cxcl15, Cxcl5, Cxcl9, Ccr1, Ccr10, Ccr2, Ccr3, Ccr4, Ccr5, Ccr6, Ccr8, Cxcr2, Cxcr3, Cxcr5,Il11, Il13, Il15, Il16, Il17a, Il17b, Il17f, Il1a, Il1b, Il1rn, Il21, Il27, Il3, Il33, Il4, Il5, Il7, Il10ra, Il10rb, Il1r1,Il2rb, Il2rg, Il5ra, Il6ra, Il6st, Aimp1, Bmp2, Cd40lg, Csf1, Csf2, Csf3, Fasl, Ifng, Lta, Ltb, Mif, Nampt,Osm, Pf4,* Spp1, *Tnf, Tnfsf10, Tnfsf11, Tnfsf13, Tnfsf13b, Tnfsf4, Vegfa,* and *Tnfrsf11b.*

Cells were lysed in TNT lysis buffer (150 mM NaCl, 1% Triton X-100, 1 mM EDTA, 1 mM EGTA, 50 mM NaF, 1 mM Na₃VO₄, 1 mM PMSF, and

protease inhibitor cocktail (Sigma)) and protein amounts from cell lysates were measured using DC protein assay (Bio-Rad). Equivalent amounts of proteins were subjected to SDS–PAGE, transferred to Immobilon-P membranes (Millipore), immunoblotted with anti-ACVR1 antibody (1:1000, Sigma, #SAB3500435), anti-GAPDH antibody (1:1000, EMD Millipore, #CB1001), anti-phospho-SMAD1/5 antibody (1:1000, Cell Signaling Technology, # 9516), anti-GFP antibody (1:1000, Takara, #632381), anti-HSP90 antibody (1:1000, BioLegend, #675402), and developed with ECL (ThermoFisher Scientific). Immunoblotting with anti-HSP90 antibody or anti-GAPDH antibody was used as a loading control.

### In vitro transduction assay of rAAV serotypes
Human FOP iPSCs, BMSCs, or ASCs were plated at a density of $1 \times 10^4$ cells/well in 24-well plate and 24 h later, they were incubated with rAAV1, rAAV2, rAAV2-TM, rAAV3, rAAV4, rAAV5, rAAV6, rAAV6.2, rAAV7, rAAV8, rAAV9, rAAVrh8, rAAVrh10, rAAVrh39, or rAAVrh43 vectors packaging the *CBA-Egfp* reporter transgene at three different titers ($10^9$–$10^{11}$/mL genome copies). 48 h later, cells were washed with PBS and EGFP expression was monitored by the EVOS FL imaging system (ThermoFisher Scientific). Alternatively, cells were lysed in TNT lysis buffer and EGFP expression was assessed by immunoblotting with anti-EGFP antibody.

### Osteoblast or chondrocyte differentiation analysis
To assess extracellular matrix mineralization in AAV-treated osteoblasts, cells were washed twice with 1X phosphate-buffered saline (PBS) and fixed in 70% EtOH for 15 min at room temperature. Fixed cells were washed twice with distilled water and then stained with 2% Alizarin red solution (Sigma, #A5533) for 5 min. Cells were then washed three times with distilled water and examined for the presence of calcium deposits. Mineralization was quantified by the acetic acid extraction method[83]. For alkaline phosphatase (ALP) staining, osteoblasts were fixed with 10% neutral buffered formalin and stained with the solution containing Fast Blue (Sigma, #FBS25) and Naphthol AS-MX (Sigma, #855). Alternatively, osteoblasts were incubated with a 10-fold diluted Alamar Blue solution (Invitrogen, #DAL1100) for cell proliferation. Subsequently, cells were washed and incubated with a solution containing 6.5 mM $Na_2CO_3$, 18.5 mM $NaHCO_3$, 2 mM $MgCl_2$, and phosphatase substrate (Sigma, #S0942), and ALP activity was measured by a spectrometer (BioRad).

To assess the chondrogenic differentiation of AAV-treated chondrogenic progenitors, cells were washed with PBS and fixed with 4% glutaraldehyde for 15 min at room temperature. Fixed cells were washed with 0.1 N HCl and stained with 1% alcian blue (Sigma, #A3157) for 30 min at room temperature. After washing with 0.1 N HCl, stained proteoglycans in the extracellular matrix were detected[79].

### Next-generation sequencing (NGS)
Transcripts of *ACVR1^WT^* and *ACVR1^R206H^* in AAV-treated human FOP iPSCs were quantitated by PCR amplicons using next-generation sequencing (NGS). Briefly, $5 \times 10^{10}$ GCs of the AAV6.2 vectors carrying EGFP control, *amiR-RH6.ACVR1^opt^*, or *amiR-RH7.ACVR1^opt^* were transduced to human FOP iPSCs for 3 days. The cDNAs synthesized from total RNA were amplified using ACVR1-targeting primers and the PCR products was subjected for NGS in the Massachusetts General Hospital Center for Computational & Integrative Biology DNA Core (Boston, MA).

### Transcriptome analysis
RNA-seq samples were obtained from human FOP iPSCs treated with rAAV6.2 vector carrying EGFP control, *amiR-RH6.ACVR1^opt^*, or *amiR-RH7.ACVR1^opt^* and mapped to the human reference genome (GCF_000001405.38_GRCh38.p12) with STAR aligner (v.2.6.1b)[84,85]. After mapping, read counts were generated by using HTSeq-count (v.0.11.3)[86]. The read counts were used for a differential expression

analysis between EGFP control (ctrl) vs. *amiR-RH6.ACVR1^opt^*, EGFP control (ctrl) vs. *amiR-RH7.ACVR1^opt^* and *amiR-RH6.ACVR1^opt^* vs. *amiR-RH7.ACVR1^opt^* using DESeq2 (v.1.28.1)[87] with the ashr shrinkage estimator (v.2.2.47)[88]. Genes with statistical significance were determined as having absolute log-fold change larger than 1.5 and having a P-value less than 0.005.

### Complete blood cell count (CBC)
CBC tests were performed to evaluate cellular components in the blood of AAV-treated mice, including white blood cells (WBCs), red blood cells (RBCs), lymphocytes, monocytes, neutrophils, and platelets (PLTs). Blood drops were collected into a microtainer EDTA tube and tested within one hour at room temperature using an automated hematology analyzer (VetScan HM5, Zoetis.USA).

### Flow cytometry for immune cell population
To isolate the splenocytes from AAV-treated mice, the spleens were quickly removed under aseptic conditions, then placed on wire mesh, and gently teased with sterile forceps. Similarly, to isolate bone marrow cells, bone marrow was aspirated from mouse femurs of AAV-treated mice. Collected splenocytes and bone marrow cells were washed with 1× phosphate-buffered saline (PBS) and then incubated with RBC lysis buffer (BioLegend, #420301) for 2–5 min at room temperature to eliminate red blood cells (RBC). Further, cells were washed twice with cold fluorescence-activated cell sorting (FACS) buffer and filtered through a sterile 40 μm cell strainer before resuspending into FACS buffer and then incubated with Fc blocking buffer (BD Biosciences, #564765) for 15 min at 4 °C. After Fc receptor blocking, cells were treated with fluorochrome-labeled antibody cocktail including anti-mouse/human CD11b APC (1:100, BioLegend, #101212), anti-mouse CD45R (B220) PerCP Cy 5.5 (1:100, Tonbo, #65-0452), anti-mouse CD3 FITC (1:100, BioLegend, #100203), anti-Mouse CD11c Brilliant Violet 510 (1:100, BioLegend, #117337), anti-mouse Ly6G PE-Cy7 (1:100, BioLegend, #127618) and anti-mouse Ly6C FITC (1:100, BioLegend, #128006) in cold FACS buffer. After treatment with Ghost Dye red 780 (1:1000, Tonbo, #13-0865-T100) for live/dead cell discrimination, cells were then subjected to acquisition on a BD LSR II flow cytometer (BD Biosciences). The data were analyzed using FlowJo (v.10.1).

### Mouse models of acquired heterotopic ossification
$5 \times 10^{12}$ vector genomes vg/kg (50 μl) of rAAV6.2 or rAAV9 vectors were t.d. injected into the quadriceps muscle of 2-month-old male wild-type or *Tie2-cre;Rosa26^mCherry^* mice (C57BL/6J) using a hollow microneedle (micronjet600, NanoPass Technologies)[36] one week post-injection of rBMP2/7-matrigel and muscle injury, and 3 weeks later, radiography of hindlimbs and frozen sections of HO tissues were performed in AAV-treated mice. For muscle injury/BMP-induced HO model[37,38], blunt muscle trauma was induced by dropping an aluminum ball onto the mouse adductor muscle (right next to the femur) and a mixture of recombinant BMP2/7 (1 μg) and Matrigel (20 μl) was injected into the injured area. A subcutaneous injection of buprenorphine was provided for analgesia. Mice were euthanized and HO was assessed by radiography, microCT, and histology at 4 weeks post-injury.

### Administration of rAAVs to *Acvr1R^(R206H)KI^* FOP mice
**Trauma-induced HO.** $5 \times 10^{12}$ vg/kg (50 μl) of rAAV9 vectors carrying CRE recombinase was t.d. injected into hindlimbs of 6-week-old male *Acvr1R^(R206H)Fl^* mice and three days later, 1 μM cardiotoxin/pinch injury was employed into AAV-injected sites. Alternatively, 6-week-old female *Acvr1R^(R206H)Fl^*;Cre-ER^T2^ mice were daily treated with intraperitoneal (i.p.) injection of tamoxifen (10 mg/kg) for 5 days and 3 days later, $5 \times 10^{12}$ vg/kg (50 μl) of rAAV9 vectors carrying *mCherry, EGFP control, amiR-RH6, ACVR1^opt^*, or *amiR-RH6.ACVR1^opt^* was t.d. injected into hindlimbs. 1 μM cardiotoxin/pinch injury was employed into the

gastrocnemius muscle 3 days after AAV injection. Finally, P1 *Acvr1R*[(R206H)Fl];Cre-ER[T2] neonates were treated with 10[11] GCs (50 µl) of rAAV9 carrying *mCherry*, *EGFP control*, *amiR-RH6*, *ACVR1*[opt], or *amiR-RH6.ACVR1*[opt] via facial vein injection and 6 weeks later, mice were i.p. injected with tamoxifen (10 mg/kg), followed by 1 µM cardiotoxin/pinch injury of the gastrocnemius muscle 3 days after AAV injection. Four weeks later, *ACVR1*[R206H] and *Cre* recombinase mRNA levels and heterotopic bone mass in the gastrocnemius muscle were assessed by RT-PCR, radiography, and microCT.

**Spontaneous HO.** P1 *Acvr1*[(R206H)Fl], *Acvr1*[(R206H)Fl];PDGFRα-cre, or *Acvr1*[(R206H)Fl];PDGFRα-cre;PDGFRα-GFP neonates were treated with 10[11] GCs (50 µl) of rAAV9 carrying *mCherry*, *EGFP control*, *amiR-RH6*, *ACVR1*[opt], or *amiR-RH6.ACVR1*[opt] via facial vein injection and a full phenotypic characterization was performed until mice were 8 weeks old. Alternatively, 6-week-old female *Acvr1*[(R206H)Fl] or *Acvr1*[(R206H)Fl];Cre-ER[T2] mice were i.v. administered 5 × 10[13] vg/kg (200 µl) of rAAV9 carrying *EGFP control* or *amiR-RH6.ACVR1*[opt] 3 days after i.p. injection of tamoxifen (10 mg/kg). 12 weeks later, mRNA levels of *ACVR1*[R206H] and *ACVR1*[opt] in the liver and HO were assessed by RT-PCR, microCT, radiography, and histology, respectively.

### Clinical HO scoring
Euthanized mice were processed for whole-body radiography and microCT analyses. Each mouse was independently scored by a minimum of two researchers, blinded as to the identity of the groups, and the average score was recorded. Clinical HO incidence or severity was calculated by adding the cumulative score of HO and skeletal deformity at all targeted sites per mouse. HO lesions and skeletal deformity were scored as mild (1), moderate (2), or severe (3) based on the estimated size and fused condition at targeted sites (jaw; left/right, thoracic/cervical vertebrae, forelimb; left/right, hip, Knee; left/right, hindlimb; left/right and ankle; left/right).

### Statistical methods
All data were presented as the mean ± SD. Sample sizes were calculated on the assumption that a 30% difference in the parameters measured would be considered biologically significant with an estimate of sigma of 10–20% of the expected mean. Alpha and Beta were set to the standard values of 0.05 and 0.8, respectively. No animals or samples were excluded from the analysis and animals were randomized to treatment versus control groups, where applicable. For relevant data analysis, where relevant, we first performed the Shapiro–Wilk normality test for checking the normal distributions of the groups. If normality tests passed, a two-tailed, unpaired Student's *t*-test was used and if normality tests failed, Mann–Whitney tests were used for the comparisons between the two groups. For the comparisons of three or four groups, we used one-way ANOVA if normality tests passed, followed by Tukey's multiple comparison test for all pairs of groups. If normality tests failed, the Kruskal-Wallis test was performed and was followed by Dunn's multiple comparison test. The GraphPad PRISM software (v6.0a, La Jolla, CA) was used for statistical analysis. $P < 0.05$ was considered statistically significant. $*P < 0.05$; $**P < 0.01$; $***P < 0.001$; and $****P < 0.0001$.

### Reporting summary
Further information on research design is available in the Nature Research Reporting Summary linked to this article.

## Data availability
Data supporting the findings of this manuscript are available from the corresponding authors upon request. The RNA-seq data generated in this study have been deposited in the NCBI database under accession code GSE213579. The raw data are protected and are not available due to data privacy laws. However, the minimum dataset generated in this study, which is necessary to interpret, verify and extend the research in the article, is provided in the Supplementary Information/Source Data file. Source data are provided with this paper.

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

## Acknowledgements

We would like to thank Drs. Daniel Perrien and Silvia Corvera for pro-viding *Acvr1*(R206H)Fl mice and human ASCs, respectively, and Eunhye Son, Oksun Lee, Zhihao Chen, and Samuel Kou for technical support, and the many individuals who provided valuable reagents. We also thank Drs. Marelise Eekhoff, Dimitra Micha, Paul Yu, Jeffrey Chamberlain, David Goldhamer, Benjamin Levi, Yuji Mishina, Adam Sherman, and Danielle Kerkovich for thoughtful discussion. G.G. is supported by grants from the NIH (P01AI100263, R01NS076991, P01HD080642, R01AI12135). J.-H.S. is supported by NIH/NIAMS (R21AR077557, R01AR078230), the International FOP Association, and AAVAA Therapeutics. F.S.K. is sup-ported by the Isaac & Rose Nassau Professorship of Orthopaedic Mole-cular Medicine. E.M.S. is supported by the Cali/Weldon Professorship for FOP Research.

## Author contributions

Y.-S.Y. and J.-M.K. designed, executed, and interpreted the experiments. H.M. and J.X. designed and generated all of the AAVs. C.L. and S.C. analyzed dental/alveolar bone and immune responses in FOP mice, respectively. E.H. provided healthy and FOP patient-derived iPSCs and assisted with editing the manuscript. J.H.H. and H.H.C. performed a whole transcriptome analysis. F.S.K. and E.M.S. supervised the research and manuscript development. G.G. and J.H.S. supervised the research and prepared the manuscript. All authors revised the manuscript and approved the final draft.

## Competing interests

G.G. and J.-H.S. have submitted a patent application concerning the methodology described in this study. G.G. and J.-H.S. are scientific co-founders of AAVAA Therapeutics and hold equity in this company. G.G. is also a scientific co-founder of Voyager Therapeutics and Aspa Ther-apeutics and holds equity in these companies. G.G. is an inventor on patents with potential royalties licensed to Voyager Therapeutics, Aspa Therapeutics Inc., and other biopharmaceutical companies. E.C.H. serves in a volunteer capacity on the registry advisory board of the IFOPA; on the International Clinical Council on FOP, and on the Fibrous Dysplasia Foundation Medical Advisory Board. E.C.H. received prior research support through his institution from Regeneron Pharmaceu-ticals. E.C.H. receives clinical trials research support through his insti-tution from Clementia, an Ipsen company. F.S.K. is the founder and past-President of the International Clinical Council (ICC) on FOP. F.S.K. serves in a volunteer capacity on the registry advisory board of the IFOPA. F.S.K. is an investigator on clinical trials sponsored by Clementia, an Ipsen company, and by Regeneron Pharmaceuticals. E.M.S. serves in a volunteer capacity as a research advisor to the IFOPA. These pose no competing interests for this study. Other authors declare no competing interests.

## Additional information

[1]Department of Medicine/Division of Rheumatology, UMass Chan Medical School, Worcester, MA, USA. [2]Horae Gene Therapy Center, UMass Chan Medical School, Worcester, MA, USA. [3]Department of Microbiology and Physiological Systems, UMass Chan Medical School, Worcester, MA, USA. [4]Viral Vector Core, UMass Chan Medical School, Worcester, MA, USA. [5]Division of Endocrinology and Metabolism, Department of Medicine; the Institute for Human Genetics; the Program in Craniofacial Biology; and the Eli and Edyth Broad Institute of Regeneration Medicine, University of California-San Francisco, San Francisco, CA, USA. [6]Department of Mathematical Sciences, Korea Advanced Institute of Science and Technology, Daejeon, Republic of Korea. [7]Department of Orthopaedic Surgery, The Perelman School of Medicine at the University of Pennsylvania, Philadelphia, PA, USA. [8]Department of Genetics, The Perelman School of Medicine at the University of Pennsylvania, Philadelphia, PA, USA. [9]The Center for Research in FOP and Related Disorders, The Perelman School of Medicine at the University of Pennsylvania, Philadelphia, PA, USA. [10]Department of Medicine, The Perelman School of Medicine at the University of Pennsylvania, Philadelphia, PA, USA. [11]Li Weibo Institute for Rare Diseases Research, UMass Chan Medical School, Worcester, MA, USA. [12]These authors contributed equally: Yeon-Suk Yang, Jung-Min Kim, Jun Xie. ✉e-mail: guangping.gao@umassmed.edu; jaehyuck.shim@umassmed.edu

