## [Peer Review File · Nature Communications]

Reviewer comments, initial review –

Reviewer #1 (Remarks to the Author):

The authors conducted an extremely interesting in-depth study to identify the best strategy to find a genetic therapy for Fibrodysplasia ossificans progressive (FOP). They identified recombinant adeno-associated virus 9 (rAAV9) as the most effective serotype for transduction of the major cells-of-origin of HO in soft tissues and found that the combination of gene replacement and silencing (ACVR1OPT;amiR-ACVR1R206H) ablated aberrant Activin A signaling and chondrogenic and osteogenic differentiation of mouse skeletal cells harboring a conditional knock-in allele of human ACVR1R206H (Acvr1(R206H)KI) and human FOP patient-derived induced pluripotent stem cells (iPSCs). In Acvr1(R206H) mice, trauma-induced HO was also markedly reduced when treated locally in early adulthood or systemically at birth.

This is the first study focusing on a genetic therapy of FOP, for which there is currently no therapy worldwide.

This is an extremely complicated disease and the authors have achieved impressive results that give hope for the future.

This manuscript is very innovative, original, the methods are well described, the results are of great importance to the field of FOP, I would certainly consider it for publication. The videos are also very interesting.

There are a number of questions that require clarification or explanation, which are summarized below.

1. Page 8 line 151: Have the authors also considered using transdifferentiation of dermal fibroblast cultures instead of FOP iPSCs, thus preventing the loss of valuable characteristics?
2. Page 13 line 281: chondrocyte lineage cells in growth plates did not appear to be rAAV9-transducible in experiments with Acvr1(R206H)KI mice, but did the authors show the same for knee cartilage?
3. Page 15 line 330: Here is stated "administration of rAAV9.amiR-RH6.ACVR1opt at birth prevented trauma-induced development of heterotopic endochondral ossification in Acvr1(R206H)KI FOP mice but showed minimal effects on early post-traumatic injury responses"
Does this mean that it has to be given more often or that cells other than FAP play a role during the already ongoing HO process? Could the authors perhaps bring their opinion about this topic into the discussion?
4. Page 18 line 422: Can the authors explain why HO formation of the cervical spine appeared to specifically occur after systemic administration
5. Page 19 line 450: It is intriguing why no effect of the administration on the immune system is seen. Can the authors indicate whether it has to do with the type of vector?
6. An article has recently been published in which the possibility of a study such as the one you have carried out is discussed. You may consider including this in the references.

Reviewer #2 (Remarks to the Author):

The manuscript presented by Yang and colleagues clearly shows the possibility to use a combination of gene replacement and inhibition to address heterotopic ossification in human and mouse cells and in vivo in rodents. The manuscript is well written and the data presented are robust. The authors clearly showed proof of concept.

The authors have to address the following comments:

1. The mouse models used have to be described in detail in the method section. For example, the different CRE used and whether they are inducible or not as well as the ACVR1 mutant that has the mutation in the mouse sequence, and this information was not easily found.
2. Please comment on the fact that you selected your artificial miRNA on a human sequence and that sequence was effective also on the mouse sequence. How similar are the target sequences in the two species?
3. One potential limitation of this work is that the use of inducible models not recapitulating the human disease and AAV treatment at different ages failed to demonstrate efficacy in a clinically relevant context. Could you please discuss this point in light of the potential target patient population?
4. The authors clearly state that rAAV has high transduction efficacy in skeletal muscle and skeleton. Please comment on the adverse events reported when doses efficiently targeting those tissues were applied in patients. This is a clear limitation of the clinical translation. Are there alternatives to target those tissues with lower vector doses?

Reviewer #1:

The authors conducted an extremely interesting in-depth study to identify the best strategy to find a genetic therapy for Fibrodysplasia ossificans progressiva (FOP). They identified recombinant adeno-associated virus 9 (rAAV9) as the most effective serotype for transduction of the major cells-of-origin of HO in soft tissues and found that the combination of gene replacement and silencing (ACVR1OPT;amiR-ACVR1R206H) ablated aberrant Activin A signaling and chondrogenic and osteogenic differentiation of mouse skeletal cells harboring a conditional knock-in allele of human ACVR1R206H (Acvr1(R206H)KI) and human FOP patient-derived induced pluripotent stem cells (iPSCs). In Acvr1(R206H) mice, trauma-induced HO was also markedly reduced when treated locally in early adulthood or systemically at birth.

This is the first study focusing on a genetic therapy of FOP, for which there is currently no therapy worldwide. This is an extremely complicated disease and the authors have achieved impressive results that give hope for the future. This manuscript is very innovative, original, the methods are well described, the results are of great importance to the field of FOP, I would certainly consider it for publication. The videos are also very interesting.

- We thank the reviewer for summarizing and highlighting the significance of our manuscript. We agree that these findings will pave the way for treating HO in FOP patients. We believe that addressing these comments has significantly improved the revised manuscript.

1. Page 8 line 151: Have the authors also considered using transdifferentiation of dermal fibroblast cultures instead of FOP iPSCs, thus preventing the loss of valuable characteristics?

- We thank the reviewer for suggesting this approach. The main purpose of the FOP iPSCs experiments was to demonstrate that the AAV targeting of *ACVR1^{R206H}* could result in normalization of gene expression levels and reduction of osteogenic markers using a multipotent cell type as a starting point. FOP iPSCs were previously shown to have increased mineralization and osteogenic gene expression (PMID 24321451); thus, the changes described in **Figure 2** and **Extended Data Figure 4** demonstrate these findings.

- Osteogenic transdifferentiation of dermal fibroblasts from patients with FOP have been previously reported, using retroviral transduction of osteogenic transcription factors and reprogramming factors, human platelet lysate, or ascorbic acid, beta glycerol phosphate, and dexamethasone (PMID 26769004, 28705683). We believe that the use of our established, highly-characterized, episomal, iPSC model, which has now been used successfully in multiple studies (PMID: 24321451, 35442931, 34755602, 34311122, 27530160), provides several key advantages for our study: The iPSCs used in this study have no retroviral reprogramming factors and do not induce cellular differentiation by overexpression of master transcription factors, both of which could over-ride cellular differentiation signals modified by *ACVR1^{R206H}*. In addition, the use of human platelet lysate is an undirected differentiation process, making the final cell lineages difficult to interpret. The ascorbic acid/beta glycerol phosphate/dexamethasone method used for dermal fibroblasts is similar to what we used with the iPSCs, but the use of iPSCs provides a key advantage of starting from a cell type capable of differentiation into multiple skeletal lineages (which could be detected using NGS), rather than starting from a stable cell type that may have limited differentiation capacity.

- Currently, we are also testing the ability of rAAV9.*amiR-RH6.ACVR1^{opt}* to suppress osteogenic differentiation of periodontal ligament fibroblasts derived from human FOP patients.

2. Page 13 line 281: chondrocyte lineage cells in growth plates did not appear to be rAAV9-transducible in experiments with Acvr1(R206H)KI mice, but did the authors show the same for knee cartilage?

- We thank the reviewer for pointing this out. A new fluorescence microscopy data showing that chondrocyte lineage cells in articular cartilage are not rAAV9-transducible was added to **Extended Data Figure 8c (top panel)**.

3. Page 15 line 330: Here is stated "administration of rAAV9.amiR-RH6.ACVR1^{opt} at birth prevented trauma-induced development of heterotopic endochondral ossification in *Acvr1*(R206H)KI FOP mice but showed minimal effects on early post-traumatic injury responses" Does this mean that it has to be given more often or that cells other than FAP play a role during the already ongoing HO process? Could the authors perhaps bring their opinion about this topic into the discussion?

- We thank the reviewer for raising these questions. Tissue-specific tropism of rAAV9 serotype might be responsible for minimal effects of rAAV9.amiR-RH6.ACVR1^{opt} on early post-traumatic injury responses. Our present and previous studies (**Figure 3** and **Extended Data Figure 3**, PMID: 32405514) and others (PMID: 35871478) demonstrated that rAAV9 serotype is effective for transduction of PDGFR α ⁺ or Tie2⁺ FAP-lineage cells, osteoblast-lineage cells (osteoprogenitors, mature osteoblasts, osteocytes), and myoblast-lineage cells (myogenic progenitors, myoblasts, myocytes), but not fibroblasts and immune cells (monocytes, macrophages, dendritic cells, neutrophils, and T and B lymphocytes). These results suggest that systemically delivered *amiR-RH6.ACVR1^{opt}* mainly impacts on rAAV9-transducible cells, including FAP-lineage cells, osteoblast-lineage cells, and myoblast-lineage cells in injured sites, resulting in suppression of chondrogenesis and osteogenesis while facilitating muscle regeneration. However, *amiR-RH6.ACVR1^{opt}* is likely to be dispensable for trauma-induced inflammation, fibroproliferation, and muscle damage (early post-traumatic injury responses) due to low transduction efficiency to fibroblasts and immune cells. This was added to the result section of the revised manuscript.

4. Page 18 line 422: Can the authors explain why HO formation of the cervical spine appeared to specifically occur after systemic administration?

- We thank the reviewer for pointing this out. As described in a previous study (PMID: 26333933), 6-week-old *Acvr1*^{(R206H)Fl};Cre-ER^{T2} mice develop spontaneous HO at multiple anatomical locations, including the cervical spine, hips and knees, 10-12 weeks post-injection of tamoxifen. These HO phenotypes might not be affected by IV injection of rAAV vectors.

5. Page 19 line 450: It is intriguing why no effect of the administration on the immune system is seen. Can the authors indicate whether it has to do with the type of vector?

- We thank the reviewer for raising this interesting question. From humoral responses point of view, since mouse is not a natural host of primate AAVs, there is little to no pre-existing neutralizing antibodies against any AAV serotypes and no negative impact on the first administration of mice. However, it will trigger the production of AAV neutralization antibodies, preventing redosing of AAVs. In terms of cellular responses, among the commonly used viral vectors, AAV is the only viral vector that elicits little to no T cell responses to AAV transduction in mice with a few exceptions for some highly immunogenic transgenes such as ovalbumin.

6. An article has recently been published in which the possibility of a study such as the one you have carried out is discussed. You may consider including this in the references.

- As suggested by the reviewer, the reference, “Gene Therapy for Fibrodysplasia Ossificans Progressiva (FOP): feasibility and obstacles” in Human Gene Therapy (PMID: 35502479), was added to the revised manuscript.

Reviewer #2:

The manuscript presented by Yang and colleagues clearly shows the possibility to use a combination of gene replacement and inhibition to address heterotopic ossification in human and mouse cells and in vivo in rodents. The manuscript is well written and the data presented are robust. The authors clearly showed proof of concept.

- We thank the reviewer for the positive comments on the significance of AAV-mediated gene therapy for FOP. We appreciate the reviewer's constructive suggestions and believe that addressing these points has strengthened the manuscript.

The authors have to address the following comments:

1. The mouse models used have to be described in detail in the method section. For example, the different CRE used and whether they are inducible or not as well as the ACVR1 mutant that has the mutation in the mouse sequence, and this information was not easily found.

- As suggested by the reviewer, detail description of the mouse models used in this study was added to the revised manuscript. Additionally, the targeting construct to generate mice harboring a conditional *Acvr1*^{R206H} knock-in allele (*Acvr1*^{(R206H)Fl}) was added to the revised manuscript (**Extended Data Fig. 7c**).

2. Please comment on the fact that you selected your artificial miRNA on a human sequence and that sequence was effective also on the mouse sequence. How similar are the target sequences in the two species?

- We thank the reviewer for raising this interesting point. We designed 12 artificial miRNAs targeting human *ACVR1*^{R206H} mRNA based on human *ACVR1* sequences and many of them showed knockdown effects on the expression of both *ACVR1*^{WT} and *ACVR1*^{R206H} mRNA in different degrees. Additionally, these artificial miRNAs have the possibility to affect mouse *Acvr1* expression, as mouse and human *Acvr1* sequences share 14 common sequences (GTGGCTC**G**CCAGAT, c617G>A;p.R206H) out of 21 target sequences. However, *amiR-RH6.ACVR1*^{opt}-expressing liver showed a significant decrease in human *ACVR1*^{R206H} expression without any alteration in mouse *Acvr1* expression (**Figure 6a** and **Extended Data Figure 12c**), suggesting that *amiR-RH6* is specific to silence the expression of human *ACVR1*^{R206H}.

3. One potential limitation of this work is that the use of inducible models not recapitulating the human disease and AAV treatment at different ages failed to demonstrate efficacy in a clinically relevant context. Could you please discuss this point in light of the potential target patient population?

- We agree with the reviewer's concern that our FOP mouse models have limitations (*Acvr1*^{(R206H)Fl};Cre-ER^{T2} mice; tamoxifen-induced expression of *ACVR1*^{R206H}, *Acvr1*^{(R206H)Fl};PDGFR α -Cre mice; localized expression of *ACVR1*^{R206H} in PDGFR α ⁺ cells) and do not recapitulate the full spectrum of human FOP phenotypes in and out of the skeleton. Additionally, our therapeutic strategy using the AAV vectors mainly focuses on preventing trauma-induced HO. Although long-term organism wide gene therapy is our ultimate goal, our findings in this study more likely represent a use-scenario for local therapy. We would speculate that flare-based treatments using transdermal injection of rAAV9.*amiR-RH6.ACVR1*^{opt} may be useful for suppressing trauma-induced HO in the early stages of a flare-up or potentially in patients undergoing surgery. These limitations and possible uses have been added to the discussion section of the revised manuscript.

4. The authors clearly state that rAAV has high transduction efficacy in skeletal muscle and skeleton. Please comment on the adverse events reported when doses efficiently targeting those tissues were applied in patients. This is a clear limitation of the clinical translation. Are there alternatives to target those tissues with lower vector doses?

- We thank the reviewer for raising this important point. Recently, an rAAV9 vector carrying SMN1 (rAAV9.SMN1, Zolgensma) was FDA-approved as a gene therapy to treat spinal muscular atrophy. Systemic infusion of a single high dose of Zolgensma (2×10^{14} vg/kg) has been proven therapeutically effective and safe in clinical trials. However, high transduction efficacy of systemically delivered rAAV9 vectors to the diseased liver (PMID:32777938) has been known to cause untoward side effect, acute serious liver injury and acute liver failure. Similar to Zolgensma, our gene therapy for FOP utilizes systemic infusion of rAAV9 capsid to deliver *amiR-RH6.ACVR1^{opt}* but at a much lower dose for targeting skeletal muscle and skeleton instead of the motor neuron. Based on our preclinical proof-of-concept gene therapy study in mouse models of FOP, AAV-mediated gene therapy showed robust therapeutic effects at $2 \sim 5 \times 10^{13}$ vg/kg, which is 4-10 fold lower than a FDA-approved dose of Zolgensma (2×10^{14} vg/kg). However, to minimize potential side effects by systemically delivered rAAV9 vectors, further vector improvements using endogenous miRNA-mediated liver-detargeting and/or capsid modification have been made to limit *amiR-RH6.ACVR1^{opt}* expression in the liver and/or to specifically deliver *amiR-RH6.ACVR1^{opt}* to HO-causing cells, respectively.

- Our results demonstrated that a single dose ($2 \sim 5 \times 10^{12}$ vg/kg) of rAAV9.*amiR-RH6.ACVR1^{opt}* via transdermal injection to the skeletal muscle was highly effective in suppressing trauma-induced HO (**Figure 3i and j**). We anticipate that direct transdermal injection of rAAV9.*amiR-RH6.ACVR1^{opt}* at 5×10^{12} vg/kg (i.e. 1/40 of IV dose of Zolgensma) to the flare-up lesions in the skeletal muscle/skeleton is also clinically translatable for both efficacy and safety.

- This was added to the discussion section of the revised manuscript.

Reviewer comments, further review –

Reviewer #1 (Remarks to the Author):

I agree with the answers and changes made by the authors.
It is a very interesting and innovative article that is certainly important to publish

Reviewer #2 (Remarks to the Author):

The authors addressed all my concerns

REVIEWERS' COMMENTS

Reviewer #1 (Remarks to the Author):

I agree with the answers and changes made by the authors.
It is a very interesting and innovative article that is certainly important to publish
: Completed

Reviewer #2 (Remarks to the Author):

The authors addressed all my concerns
: Completed